# Ultrafast structural dynamics of carbon–carbon single-bond rotation in transient radical species at non-equilibrium

Seonggon Lee [1], Hosung Ki [1], Donghwan Im[1,2], Jungmin Kim [1,2], Yunbeom Lee[1], Jain Gu [1,2], Alekos Segalina[1], Jun Heo [1], Yongjun Cha [1,2], Kyung Won Lee[1,2], Doyeong Kim [1,2], Jeongho Kim [3], Rory Ma[4], Jae Hyuk Lee [4] & Hyotcherl Ihee [1,2] ✉

Bond rotation is an important phenomenon governing the fate of reactions. In particular, heterogeneously substituted ethane derivatives provide distinct structural conformations around the bond, empowering them as ideal systems for studying the rotation along carbon-containing single bonds. However, structural dynamics of ultrafast single-bond rotation, especially along C–C• bonds, have remained elusive as tracking the detailed changes in structural parameters during the rotational isomerization is challenging with conventional spectroscopic tools. Here, we employ femtosecond time-resolved X-ray liquidography to visualize the rotational isomerization between *anti* and *gauche* conformers of tetrafluoroiodoethyl radical ($C_2F_4I•$) and 1,2-tetrafluorodiiodoethane ($C_2F_4I_2$), simultaneously. The TRXL data captures perturbations in conformer ratios and structures of each reacting species, revealing that the rotational isomerization of $C_2F_4I•$ and $C_2F_4I_2$ follows *anti*-to-*gauche* and *gauche*-to-*anti* paths with time constants of 1.2 ps and 26 ps, respectively. These findings also align with the computational predictions. This work offers an atomic-level insight into the kinetics and structural dynamics of single-bond rotation.

The internal rotation of molecules is foundational in chemistry, governing the reactivity, stereochemistry, and fate of reactions[1]. In particular, the rotation of single bonds containing carbon atoms plays an indispensable role in synthetic and biological chemistry. The rotational degree of freedom around carbon–carbon (C–C) single bonds provides flexibility to the three-dimensional orientation of atoms in a molecule, giving rise to the concept of conformational isomerism. The conformational isomerism determines the stereochemistry of the products in some key reactions[2,3] and affects the three-dimensional structure of proteins by altering interactions between residues[4–7].

Therefore, understanding the chemical origins, structural dynamics, and potentials of manipulation for the internal single-bond rotation processes is vital for synthesizing complex molecules with desired functionalities, improving the adaptability of proteins, and facilitating the binding of substrates[8–13].

The conformer behaviors involved in single-bond rotation are decisively determined by *dynamic equilibrium*, where the forward and reverse transitions have equal rates. The dynamics of equilibration occurring between multiple conformers are settled by electronic, vibrational, and rotational eigenstates, as well as the conformational

[1]Center for Advanced Reaction Dynamics, Institute for Basic Science (IBS), Daejeon 34141, Republic of Korea. [2]Department of Chemistry and KI for the BioCentury, Korea Advanced Institute of Science and Technology (KAIST), Daejeon 34141, Republic of Korea. [3]Department of Chemistry, Inha University, 100 Inha-ro, Michuhol-gu, Incheon 22212, Republic of Korea. [4]Pohang Accelerator Laboratory, Pohang, Gyeongbuk 37673, Republic of Korea. ✉e-mail: hyotcherl.ihee@kaist.ac.kr

landscape of a molecule. The simplest molecule featuring rotational isomerism along C−C bond is ethane, where the rotation occurs between two conformers: *staggered* and *eclipsed*[1,14,15]. Accordingly, ethane and its derivatives have gathered spotlights as models for studying single-bond rotation in optical spectroscopy[16–20] and computational chemistry[1,14,15,21–23]. However, non-substituted ethane has drawbacks as a candidate system to resolve the structural aspects of single-bond rotation. Notable experimental techniques directly sensitive to molecular structures, such as scattering or diffraction, are not very sensitive to extremely light atoms like hydrogens. Hence, the lack of structural sensitivity for the six hydrogen atoms around the C−C bond makes it challenging to retrieve the structural snapshots of bond rotation in ethane (see the sensitivity plot in Supplementary Fig. 2 [24]).

On the other hand, the two *staggered* conformers in heterogeneously substituted ethane derivatives (*anti* and *gauche*) render alternative avenues to study the C−C single-bond rotation, as they possess distinctive substituents which serve as fingerprints for monitoring single-bond rotation from structural perspectives. Early theoretical studies have shown that the rotational isomerization of *n*-butane, one of the simplest ethane derivatives, occurs with time constants ranging from 1 ps to 100 ps at room temperature under varying chemical environments[22,23]. A two-dimensional infrared vibrational echo spectroscopy study on another ethane derivative (1-fluoro-2-isocyanato-ethane) revealed a time constant of 43 ps for the C−C bond rotation[16]. Subsequently, time-resolved infrared and nuclear magnetic resonance spectroscopy studies reported time constants ranging from 6.3 ps (along C−C bond)[17], 47 ps (along C−C• bond)[18], and 3 ps to 27 ps (along C−N bond)[19] under various steric environments around the rotating bond. Nevertheless, time-resolved optical spectroscopy and nuclear magnetic resonance commonly highlight either ultrafast temporal resolution or atomic-level structural sensitivity, but hardly feature both simultaneously[25,26]. Meanwhile, an accurate retrieval of structural dynamics requires a simultaneous interpretation of both realms. Accordingly, the ultrafast evolution of atomic-level molecular structures during single-bond rotation remains largely unexplored in solutions.

Time-resolved X-ray liquidography (TRXL), or time-resolved X-ray solution scattering (TRXSS), provides direct and detailed information on structural dynamics[27–30], complementing the insights obtained from optical spectroscopy. To fully leverage this structural sensitivity, we chose a derivative with three hydrogen atoms at each carbon site of ethane replaced with one iodine and two fluorine atoms. As iodine and fluorine atoms have larger scattering form factors, the structural dynamics of single-bond rotation can be directly visualized by tracing how these substituted atoms move. The photodynamics of this molecule, 1,2-tetrafluorodiiodoethane ($C_2F_4I_2$), start with the dissociation of one carbon–halogen bond, thereby breaking the symmetry between the two carbon atoms (one carbon becomes $CF_2I$ and the other becomes $CF_2$•) and making the rotational isomerization along C−C• bond in $C_2F_4I$• easily resolvable.

Each of $C_2F_4I_2$ and $C_2F_4I$• has two stable conformers, *anti* and *gauche*, whose population ratio can change after excitation. In fact, a study using density functional theory (DFT) calculations suggested that *anti* and *gauche* conformers of $C_2F_4I_2$ exhibit different absorption cross-sections. For example, in one computational study[31,32], the relative *anti*-to-*gauche* ratio of the excitation probability to the electronic states accessible by 267 nm photon ($^1B_u$ in *anti*, $^1B(1)$ in *gauche*) ranges from 80:20 to 89:11, indicating that the excitation of *anti*-$C_2F_4I_2$ dominates over *gauche*-$C_2F_4I_2$. These excitation ratios from $C_2F_4I_2$ to $C_2F_4I$• are determined by the electronic properties of each $C_2F_4I_2$ conformer. Meanwhile, the dynamic equilibrium ratios of $C_2F_4I_2$ and $C_2F_4I$• are determined by the thermodynamic properties, implying that the excitation ratio need not to be the same as the equilibrium ratio. These considerations suggest the possibility to trigger and capture the ultrafast single-bond rotation along C−C and C−C• bonds by observing

the re-equilibration of photoinduced perturbations in *anti*-to-*gauche* ratios of $C_2F_4I_2$ and $C_2F_4I$• (Fig. 1). Although the photodissociation of $C_2F_4I_2$ has been extensively studied in gas[33–35] and solution phases[36,37], tracking the time-dependent *anti*-to-*gauche* ratios during the rotational dynamics has not been fully achieved.

In this work, we conduct a femtosecond TRXL (fs-TRXL) experiment to investigate the structural dynamics of single-bond rotation occurring within 100 ps after photoexcitation. The experimental setup for fs-TRXL is illustrated in Supplementary Fig. 1. Since the fraction of molecules participating in the rotational isomerization is far smaller than those undergoing the dissociation path, we utilize ultrabright X-ray pulses from an X-ray free-electron laser facility to capture subtle transient signals hidden beneath the dissociation dynamics. The high sensitivity of TRXL on heavy atom positions (see Supplementary Fig. 2 for the sensitivity plot[24]) enables the accurate retrieval of structural motions during rotational isomerization along single bonds (C−C and C−C•) with sub-Å precisions. Moreover, the femtosecond resolution of fs-TRXL enables the capture of sub-ps kinetics, which is particularly relevant to the rotation around C−C• bonds. By analyzing the fs-TRXL data in both reciprocal and real space, we successfully reveal the time-dependent structural dynamics, setting the stage for a detailed exploration of the single-bond rotation.

## Results and discussion
### Femtosecond TRXL of $C_2F_4I_2$
To investigate the photoinduced structural dynamics of rotation and dissociation along carbon-containing single bonds, we measured time-resolved two-dimensional X-ray scattering images of the $C_2F_4I_2$ solution. These scattering images were decomposed into one-dimensional isotropic and anisotropic curves by utilizing the anisotropic signal decomposition method (see section 1-2 of Supplementary Information (SI) for details)[38]. The one-dimensional isotropic X-ray scattering curves ($S_0$) collected after the photoexcitation were subtracted by those measured without the laser to obtain difference isotropic X-ray scattering curves. The difference isotropic X-ray scattering curves originate from the atomic pairs (1) within the solute, (2) between the solute and solvent (also known as the cage term) and (3) between the solvent molecules. Here, only the first two contributions (solute-only and cage) encrypt the structural dynamics of the solute and are termed "solute-related". We quantitatively eliminated the contributions of (3), $\Delta S_0^{heat}(q, t)$, which originate from solvent heating (thus labeled as the solvent term), from the difference isotropic X-ray scattering curves through the projection to extract the perpendicular component (PEPC) method[39], leaving only the solute-related data ($q\Delta S_0(q, t)$, see Fig. 2a). The time-dependent changes of $q\Delta S_0(q, t)$ are indicative for the solute-related structural dynamics. We constructed a comprehensive structural dynamic model, whose details are discussed throughout the following paragraphs, to disclose the origins of these features. The simulated TRXL curves ($q\Delta S_0'(q, t)$, see Fig. 2b) coincide well with the experimental curves.

### Rotational isomerization along C−C and C−C• bonds
To extract the kinetic information of the solute-related dynamics, we applied singular value decomposition (SVD)[40] on the TRXL curves. SVD decomposes the data into a product of left singular vectors (LSV($q$)), singular values (S), and right singular vectors (RSV($t$)). Here, LSV($q$) contains the representative signals in $q$-space that encode the structural information of each chemical species, RSV($t$) describes their temporal evolution, and S quantifies the relative contribution of each rank to the data (see Methods and section 1-6 of SI for details). From SVD (in Supplementary Fig. 5), we identified a total of five major components participating in the photodynamics of $C_2F_4I_2$. To extract the solute-related kinetics, the corresponding five RSVs were globally fitted with a sum of four exponential functions, characterized by the time constants of $130 \pm 50$ fs, $1.2 \pm 0.4$ ps, $26 \pm 2.4$ ps, and $292 \pm 136$ ps,

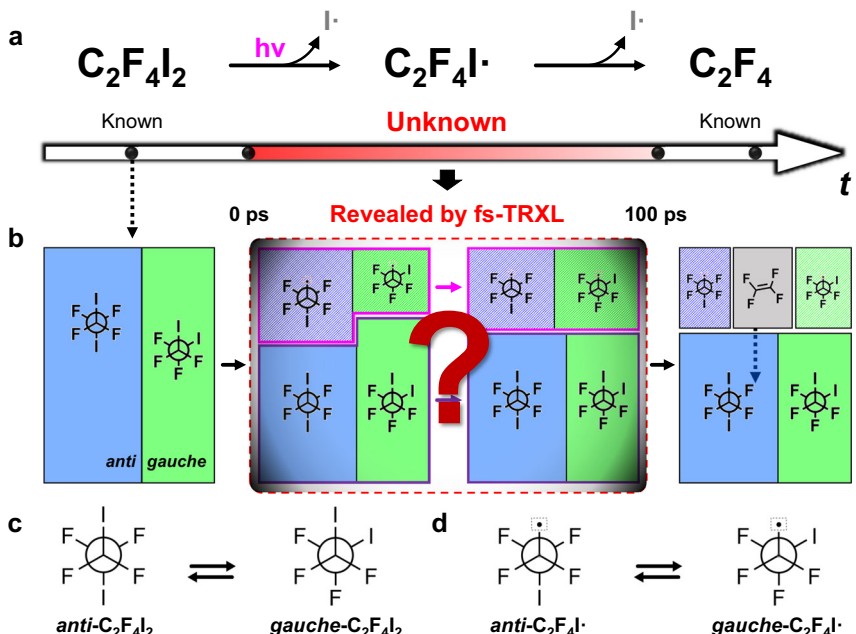

**Fig. 1 | Photoinduced structural dynamics of 1,2-diiodotetrafluoroethane (C₂F₄I₂).** **a** Two-step photoinduced dissociation pathway of $C_2F_4I_2$. It is known that $C_2F_4I_2$ absorbs an ultraviolet photon at 267 nm to undergo the two-step I• dissociation to form the radical intermediate ($C_2F_4I\bullet$) and the final product ($C_2F_4$). In cyclohexane, the secondary dissociation from $C_2F_4I\bullet$ to $C_2F_4$ is known to occur with a time constant of 292 ps, but the photodynamics before 100 ps remains unknown (emphasized with red bar). **b** The unknown ultrafast structural dynamics of $C_2F_4I_2$, depicting possible species involved in the reaction. Before photoexcitation, the $C_2F_4I_2$ molecules present as a mixture of *anti* (blue) and *gauche* (green) conformers. Photoinduced primary iodine dissociation depletes each conformer by different portions (purple-bordered box) depending on absorption cross-section and quantum yields. This also leaves the remaining conformational mixture $C_2F_4I_2$ whose *anti*-to-*gauche* ratio is perturbed from the dynamic equilibrium (magenta-bordered box). **c**, **d** The rotational isomerization of (**c**) $C_2F_4I_2$ and (**d**) $C_2F_4I\bullet$. These perturbed *anti*-to-*gauche* ratios recover their dynamic equilibrium through the rotational isomerization of (**c**) $C_2F_4I_2$ and (**d**) $C_2F_4I\bullet$, while preserving the overall concentration depicted by the area of solid-bordered boxes in (**b**).

convoluted with an instrument response function (IRF) with a full-width at half maximum (FWHM) of 172 ± 47 fs representing the temporal resolution of the experiment (Fig. 2c).

To assess the dynamic origins of these time constants, we analyzed the decay-associated difference scattering curves (**DADS**(q)), akin to a famous concept of decay-associated spectra in time-resolved spectroscopy. Here, each DADS except for the last one ($DADS_k(q)$) stands for the difference between scattering curves from the solution before and after the dynamics, $S_{before} - S_{after}$, occurring with the time constant of $\tau_k$, while the last DADS accounts for the difference between initial and final states, $S_{final} - S_{initial}$ (see section 3-2 of SI). The boldface indicates the matrix form; for instance, the $k^{th}$ column vector of **DADS**(q) is $DADS_k(q)$. We constructed the matrix of normalized decays ($C_{DADS}'(t)$) and extracted **DADS**(q) such that the matrix product **DADS**(q) × $C_{DADS}'(t)^T$, which equals the simulated curves ($\Delta S_0'(q, t)$), best agrees with $\Delta S_0(q, t)$. We extracted DADSs for a total of four time constants (130 fs, 1.2 ps, 26 ps, and 292 ps).

To attribute the dynamic origins to each $DADS_k(q)$, we simulated difference isotropic scattering curves for various candidate structural transitions (Supplementary Fig. 6) that could take place upon the photoexcitation of $C_2F_4I_2$. For example, $DADS_2(q)$ was compared with the simulated difference X-ray scattering curves corresponding to (i) the rotational isomerization of $C_2F_4I_2$, (ii) the rotational isomerization of $C_2F_4I\bullet$, (iii) the dissociation of two iodine atoms from $C_2F_4I_2$ to form $C_2F_4 + 2I\bullet$, and (iv) the formation of a hypothetical isomer ($I_2...C_2F_4$, suggested in one previous study[41]) from either $C_2F_4I_2$, $C_2F_4I\bullet + I\bullet$, or $C_2F_4 + 2I\bullet$. We identified the best candidates, which have the smallest reduced chi-square ($\chi_v^2$) values, to describe the four DADSs (Fig. 3a). $\chi_v^2$ is widely utilized as a statistical criterion to evaluate the fitness of models[26,27,42], whose details can be found in section 3-1 of SI. The comparison of each DADS with all possible pathways is demonstrated in Supplementary Figs. 8–10. Based on the statistical considerations,

we identified the following candidates that best describe the DADSs: (1) the primary photodissociation of an *anti* and *gauche* mixture of $C_2F_4I_2$ to a perturbed mixture of $C_2F_4I\bullet$ (for $DADS_1(q)$ with 130 fs), (2) the *anti*-to-*gauche* rotational isomerization of $C_2F_4I\bullet$ (for $DADS_2(q)$ with 1.2 ps), (3) the *gauche*-to-*anti* rotational isomerization of $C_2F_4I_2$ (for $DADS_3(q)$ with 26 ps), and (4) the secondary dissociation of $C_2F_4I\bullet$ to $C_2F_4$ and I• (for $DADS_4(q)$ with 292 ps). These "correct" pathways are summarized in Fig. 3a. We confirmed the statistical significance of each dynamic pathway by analyzing the changes in discrepancies between experimental and simulated TRXL curves when each pathway was excluded the model (Supplementary Fig. 11).

The integrated analysis of $DADS_k(q)$s with the previously assigned time constants shed light on the comprehensive structural dynamics. Firstly, upon photoexcitation, the primary I• dissociation occurs in both $C_2F_4I_2$ conformers with the apparent time constant of 130 fs ($DADS_1(q)$). We use the term "apparent" to emphasize that the ultrafast I• dissociation from $C_2F_4I_2$ in fact occurs through a complicated mechanism rather than a mere exponential kinetics. This is already hinted in the discrepancy between experimental and simulated RSVs (especially in the second RSV, see section 1-5 and Supplementary Fig. 5 of SI). We described this ultrafast temporal range by considering coherent and nonadiabatic real-time atomic motions detailed in section 5 and Supplementary Fig. 14 of SI. Nevertheless, $DADS_1(q)$ can approximate the changes in scattering curve associated with the transition from the two conformers of $C_2F_4I_2$ to the transient mixture of $C_2F_4I_2$ and $C_2F_4I\bullet$ at which the coherent and nonadiabatic atomic motions—or the transition occurring with the time constant of 130 fs in the exponential kinetics picture—complete. In other words, while we need an in-depth analysis (in section 5 of SI) to describe *how* the intricate sub-ps dynamics occurs, the changes in the scattering curve *before and after* the ultrafast dissociation are incorporated in $DADS_1(q)$. Once the dynamics related to $DADS_1$ completes, the amount

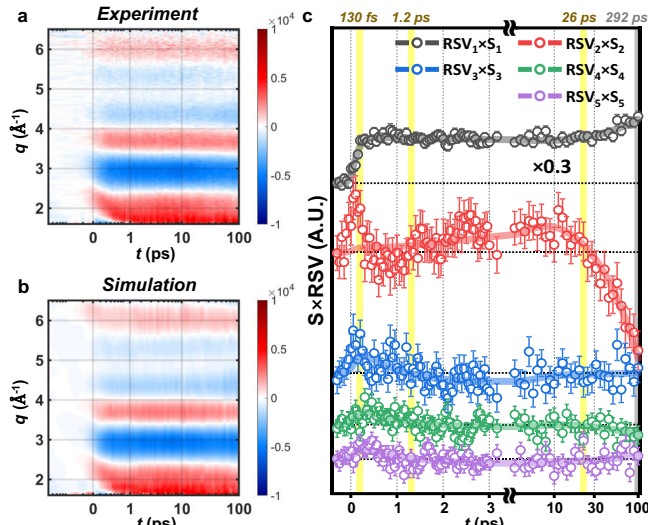

**Fig. 2 | Kinetics extracted from femtosecond time-resolved X-ray liquido-graphy (fs-TRXL) data. a** The isotropic TRXL data ($q\Delta S_0$) of $C_2F_4I_2$ in cyclohexane. **b** The simulated TRXL data, $q\Delta S_0'$. Isotropic TRXL data can be attributed to the molecular structural dynamics, such as primary and secondary dissociation of I• and rotational isomerization. **c** The five significant right singular vectors (RSVs, circles with vertical error bars) from the singular value decomposition (SVD) analysis of the experimental data. The error bars were estimated using the method described in Section 2-3 of the SI, where the LSVs in Supplementary Fig. 5a were used to obtain the time-dependent coefficients (S×RSV) as parameterized variables with standard errors. We conducted a global fit analysis with the sum of four exponentials, each with time constants of 130 fs, 1.2 ps, 26 ps (yellow shades), and 292 ps (grey shades to imply that the last dynamics happens slowly after the latest time delay), convoluted with a Gaussian IRF function with a width of 172 fs.

of dissociated *anti* and *gauche* conformers are initially transferred intact to the *anti* and *gauche* $C_2F_4I$• mixture. However, the *anti*-to-*gauche* ratios of both remaining $C_2F_4I_2$ and the newly-formed $C_2F_4I$• immediately after the primary photodissociation differ from those in dynamic equilibrium. Thus, the *anti* and *gauche* $C_2F_4I$• approach their dynamic equilibrium via *anti*-to-*gauche* isomerization along C–C• bond with the time constant of 1.2 ps (DADS$_2$($q$)). The non-dissociated $C_2F_4I_2$ mixture also returns to the dynamic equilibrium via *gauche*-to-*anti* isomerization along C–C bond with the time constant of 26 ps (DADS$_3$($q$)). Finally, secondary dissociation occurs in 30% of the equilibrated $C_2F_4I$• mixture with a time constant of 292 ps, resulting in a 70:30 mixture of $C_2F_4I$• and $C_2F_4$ (DADS$_4$($q$)), consistent with the previous ps-TRXL study[37].

In general, the observed rotational isomerization time constants of 1.2 ps and 26 ps are comparable to those reported in several previous studies, which range from a few picoseconds to tens of picoseconds. However, we note that the time constant of 1.2 ps for the C–C• bond is drastically faster than the time constant of 26 ps for the C–C bond. This discrepancy may arise from different distributions of vibrational and rotational states contributing distinctively to the pre-exponential factor, and the energetically hot nature of $C_2F_4I$• (see section 7 of SI for details). To theoretically validate the observed kinetics, we applied Rice-Ramsperger-Kassel-Marcus (RRKM) theory[43–45] to estimate the time constants for the observed rotational motions. To do this, we first calculated the shape of the potential energy surface (PES) with respect to the dihedral angle ($\varphi_{dihed}$) for $C_2F_4I_2$ and $C_2F_4I$• and identified two transition states (T1 and T2) along $\varphi_{dihed}$ (Fig. 4a and Supplementary Fig. 16). Here, it becomes evident that the rotational isomerization of both $C_2F_4I_2$ and $C_2F_4I$• proceeds through the T2 state (located between *anti* and *gauche*). To quantitatively assess the time constants, we computed the molecular

geometry, free energies of ground and excited states, vibrational frequencies, and rotational moments of inertia for the three conformers (*anti*, *gauche*, and T2) of $C_2F_4I_2$ and $C_2F_4I$•. Incorporating these data into RRKM theory provided the time constants of 20.2 ps for the *gauche*-to-T2-to-*anti* rotation of $C_2F_4I_2$ and 1.15 ps for the *anti*-to-T2-to-*gauche* rotation of $C_2F_4I$•. These estimated values show close accordance with the experimental time constants of 26 ps and 1.2 ps (see Table 1).

## Structural dynamics of rotational isomerization

To fully leverage the high sensitivity of TRXL on molecular structures, based on the sensitivity plot[24] (see Supplementary Fig. 2), we iteratively optimized four selected structural parameters–the C–I distance ($r_{CI}$), the CCI angle ($\theta_{CCI}$), the planar angle between C–I axis and FCF plane ($\theta_{pl}$), and $\varphi_{dihed}$–describing the molecular geometry of *anti*-$C_2F_4I_2$, *gauche*-$C_2F_4I_2$, *anti*-$C_2F_4I$• and *gauche*-$C_2F_4I$•. Moreover, we introduced four concentration-related parameters to account for population dynamics of excitation and equilibration. Using these structural and concentration-related parameters, the simulated DADSs (**DADS'**($q$)) were computed and superimposed with **DADS**($q$) (Fig. 3b). This rigorous analysis allows retrieving structural (Table 2) and concentration-related (Table 3) parameters for each reacting species. The signs of the first three DADSs were reversed so that the negative and positive signs match with the "depletion" and "formation" of the species. The methodological details are described in the "Structure refinement" section of Methods and section 3 of SI.

TRXL is renowned for its ability to visualize structural changes in real space with sub-Å precision. In real ($r$) space, the formation and depletion of interatomic pair distances can be more directly visualized by positive and negative signs of the oscillating $r$-space intensities. To do this, both **DADS'**($q$) and **DADS**($q$) in reciprocal space were Fourier sine transformed to the real space (**DADS'**($r$) and **DADS**($r$), see Fig. 3c). In DADS$_2$($r$), the *anti*-to-*gauche* rotation of $C_2F_4I$• is depicted by the depletions of interatomic distances in *anti*-$C_2F_4I$• ($r_{FI}^a = 3.54$ Å and 3.76 Å) and the formations in *gauche*-$C_2F_4I$• ($r_{FI}^g = 3.28$ Å and 4.25 Å). Likewise, in DADS$_3$($r$), the *gauche*-to-*anti* rotation of $C_2F_4I_2$ is expressed by the depletions of interatomic distances in *gauche*-$C_2F_4I_2$ ($r_{II}^G = 4.00$ Å, $r_{FI}^G = 2.76$ Å and 3.43 Å) and the formations in *anti*-$C_2F_4I_2$ ($r_{II}^A = 4.99$ Å, $r_{FI}^A = 2.87$ Å and 3.20 Å). DADS$_1$($r$) and DADS$_4$($r$) show a combination of formation and depletion, as both conformers of $C_2F_4I_2$ and $C_2F_4I$• contribute to the primary and secondary iodine dissociation dynamics.

## Time-dependent population dynamics revealed by TRXL

To retrieve the time-dependent concentrations of each component, we conducted the linear combination fit (LCF) analysis. From the combined analysis of SVD and DADS, we found five major species (*anti*-$C_2F_4I$•, *gauche*-$C_2F_4I$•, *anti*-$C_2F_4I_2$, *gauche*-$C_2F_4I_2$, and $C_2F_4$) contributing to $\Delta S_0(q, t)$. The LCF results (in Fig. 3e) highlight both *gauche*-to-*anti* rotation of $C_2F_4I_2$ and *anti*-to-*gauche* rotation of $C_2F_4I$• that take place with the time constant of 26 ps and 1.2 ps, respectively. These concentration profiles align well with the kinetics identified from the DADS analysis, validating our model and optimized structures.

The optimized concentration-related parameters (Table 3) and the LCF results (Fig. 3e) combined with the theoretical considerations provide a comprehensive picture of single-bond rotation dynamics. Initially, in the 60 mM $C_2F_4I_2$ solution, both conformers comprise the dynamic equilibrium with the *anti*-to-*gauche* ratio of 79($\pm$4.8):21. The 267 nm photon activates the antibonding orbitals along the C–I bond in both conformers, exciting the molecule to the Franck-Condon state. Within ultrafast time, the intersystem crossing occurs to the dissociative states (see Supplementary Fig. 15), where one C–I bond ultimately dissociates with the apparent time constant of 130 fs. The total amount of excited molecules is 6.8 mM, where 4.3 mM relaxes back to the ground state and 2.5 mM follows the dissociative path (see section

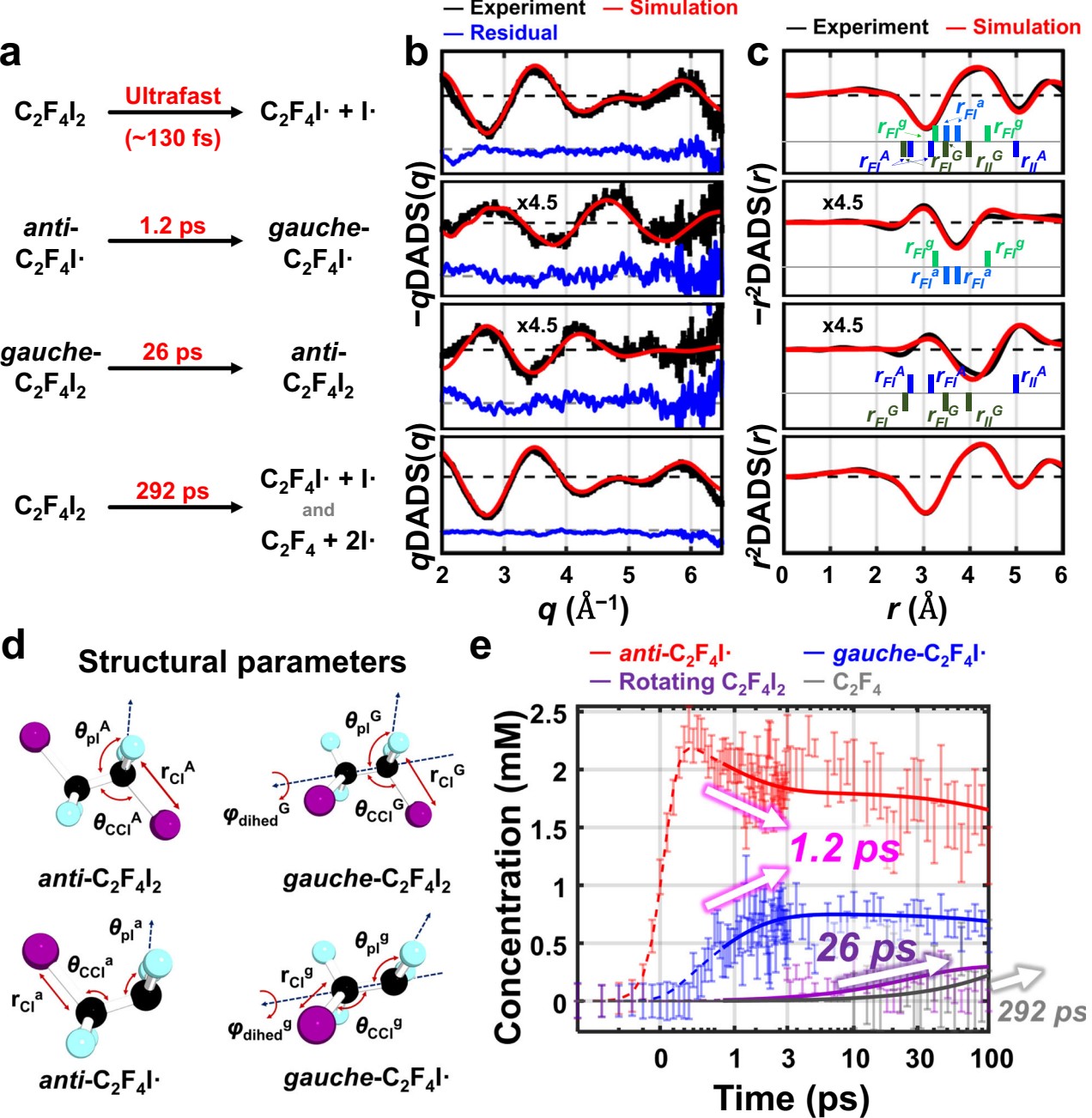

**Fig. 3 | Ultrafast structural dynamics of C−C and C−C• bond rotation. a** The reaction processes occurring in the reacting solution of $C_2F_4I_2$ within 100 ps after photoexcitation, corresponding to the time constants of 130 fs, 1.2 ps, 26 ps, and 292 ps. **b** The experimental decay-associated difference scattering curves ($q\text{DADS}(q)$, black lines) corresponding to the four time constants, superimposed with respective simulated DADSs ($q\text{DADS}'(q)$, red lines). We modeled various difference scattering curves corresponding to the candidate transition processes that can occur upon photoexcitation of $C_2F_4I_2$ (see Supplementary Figs. 8–10 for details) and identified that the four time constants are best attributable to the structural transitions shown in (**a**). The experimental error and fit residuals are indicated with vertical bars and blue lines, respectively. **c** The $r$-space representation of the experimental (black) and simulated (red) DADSs. In the 2nd DADS, interatomic distances in the *anti*-$C_2F_4I$• (light blue lines, $r_{FI}^a = 3.5$ Å and 3.8 Å) and *gauche*-$C_2F_4I$• (light green lines, $r_{FI}^g = 3.3$ Å and 4.3 Å) vanish and emerge, respectively, implying the *anti*-to-*gauche* isomerization of the radical. Likewise, in the 3rd DADS, interatomic distances in the *anti*-$C_2F_4I_2$ (blue lines, $r_{II}^A = 5.0$ Å and $r_{FI}^A = 2.9$ Å and

3.2 Å) and *gauche*-$C_2F_4I_2$ (green lines, $r_{II}^G = 4.0$ Å, $r_{FI}^G = 2.8$ Å and 3.4 Å) emerge and vanish, respectively, supporting the *gauche*-to-*anti* isomerization of $C_2F_4I_2$. We note that only the contributions of solute-related dynamics to the scattering signal are shown in (**b**) and (**c**). **d** Structural parameters of *anti*-$C_2F_4I_2$, *gauche*-$C_2F_4I_2$, *anti*-$C_2F_4I$•, and *gauche*-$C_2F_4I$• used for the iterative optimization. The simulated curves in (**b**) and (**c**) were generated using these optimized structural parameters. **e** The time-dependent concentration profiles with respect to the four components: *anti*-$C_2F_4I$• (red), *gauche*-$C_2F_4I$• (blue), the *gauche*-to-*anti* rotation of $C_2F_4I_2$ (purple), and $C_2F_4$ (gray). The error bars were estimated using the method described in section 2-3 of the SI, where the SADSs in Supplementary Fig. 12b were used to obtain the concentration profiles as parameterized variables with standard errors. We note that the third component accounts for the time-dependent concentration change of both *anti*-$C_2F_4I_2$ and *gauche*-$C_2F_4I_2$ simultaneously, as detailed in the methods section. In (**e**), the results of linear combination fit (LCF) and the least-$\chi^2$ optimization based on the kinetic modeling are shown with bars and solid lines, respectively.

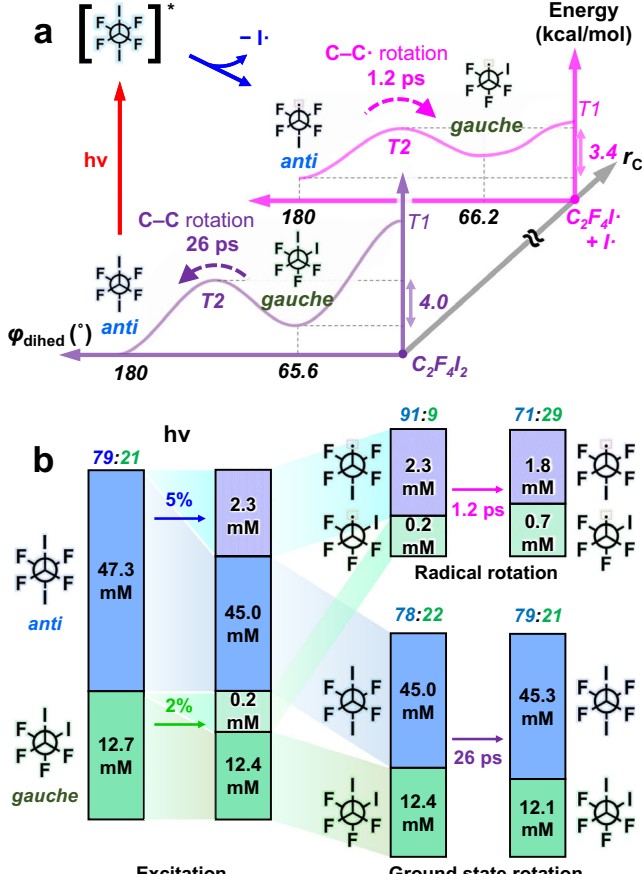

**Fig. 4 | Photodynamics of C$_2$F$_4$I$_2$ as revealed by time-resolved X-ray liquidography. a** A schematic diagram of primary iodine dissociation and rotational isomerization on the potential energy surface (PES) with respect to $r_{CI}$ and $\varphi_{dihed}$. UV photon irradiation majorly excites *anti*-C$_2$F$_4$I$_2$ molecules to the Franck–Condon region (red arrow). Thee excited molecules (denoted with an asterisk) rapidly dissociate into *anti*-C$_2$F$_4$I• and I• (blue arrow). Along the PES of C$_2$F$_4$I• (magenta coordinate), an excess amount of *anti*-C$_2$F$_4$I• molecules undergo *anti*-to-*gauche* rotational isomerization (magenta arrow), overcoming the energy barrier of 3.4 kcal/mol with a time constant of 1.2 ps. Meanwhile, along the PES of C$_2$F$_4$I$_2$ (violet coordinate), a depleted portion of *anti*-C$_2$F$_4$I$_2$ molecules re-equilibrate by *gauche*-to-*anti* rotational isomerization of C$_2$F$_4$I$_2$ (violet arrow), surpassing the energy barrier of 4.0 kcal/mol with a time constant of 26 ps. We note that, at the initial step, only the *anti*-to-*anti* excitation is depicted in the diagram for simplicity. **b** Time-dependent concentrations of *anti*-C$_2$F$_4$I$_2$, *gauche*-C$_2$F$_4$I$_2$, *anti*-C$_2$F$_4$I•, and *gauche*-C$_2$F$_4$I•. Upon photodissociation, 5% of *anti* (blue) and 2% of *gauche* (green) C$_2$F$_4$I$_2$ convert to their respective conformers of C$_2$F$_4$I•, perturbing the *anti*-to-*gauche* ratios of both C$_2$F$_4$I$_2$ (to 45.0 mM:12.4 mM = 78:22) and C$_2$F$_4$I• (to 2.3 mM:0.2 mM = 91:9). Subsequently, 0.5 mM of *anti*-C$_2$F$_4$I• converts to *gauche*-C$_2$F$_4$I• with a time constant of 1.2 ps (upper-right panel), converging to a dynamic equilibrium anti-to-gauche ratio of C$_2$F$_4$I• (to 1.8 mM:0.7 mM = 71:29). Likewise, 0.3 mM of *gauche*-C$_2$F$_4$I$_2$ isomerizes to *anti*-C$_2$F$_4$I• with a time constant of 26 ps (lower-right panel), recovering a dynamic equilibrium (to 45.3 mM:12.1 mM = 79:21). Finally, 30% of the equilibrated C$_2$F$_4$I• undergoes secondary dissociation into the product (C$_2$F$_4$) with a time constant of 292 ps, while the remaining C$_2$F$_4$I• persist up to 100 ps (as can be seen in Fig. 3a–c).

6-4 and Supplementary Fig. 17 of SI). When the C–I bond completely dissociates, 5% of *anti*-C$_2$F$_4$I$_2$ and 2% of *gauche*-C$_2$F$_4$I$_2$ convert to their corresponding C$_2$F$_4$I• conformers. At this point, the C$_2$F$_4$I• mixture contains 2.3 mM of *anti*-C$_2$F$_4$I• and 0.2 mM of *gauche*-C$_2$F$_4$I•, with an *anti*-to-*gauche* ratio of 91($\pm$11):9. Soon, the C$_2$F$_4$I• mixture reaches dynamic equilibrium with a time constant of 1.2 ps, altering the radical composition to 1.8 mM of *anti*-C$_2$F$_4$I• and 0.7 mM of *gauche*-C$_2$F$_4$I•,

leading to an equilibrium *anti*-to-*gauche* ratio of 71($\pm$8.9):29. Meanwhile, the unbalanced depletion of C$_2$F$_4$I$_2$ leaves 45.0 mM of *anti*-C$_2$F$_4$I$_2$ and 12.4 mM of *gauche*-C$_2$F$_4$I$_2$, modifying the *anti*-to-*gauche* ratio of C$_2$F$_4$I$_2$ to 78($\pm$3.6):22. This perturbed mixture returns to dynamic equilibrium with a time constant of 26 ps, as captured by the conversion of 0.3 mM of *gauche*-C$_2$F$_4$I$_2$ to *anti*-C$_2$F$_4$I$_2$. The relatively larger *gauche* fraction of C$_2$F$_4$I• compared to C$_2$F$_4$I$_2$ at dynamic equilibrium implies that the free energy difference between the two conformers is smaller in C$_2$F$_4$I• than in C$_2$F$_4$I$_2$ (see Supplementary Fig. 16). Finally, 30% of the equilibrated C$_2$F$_4$I• undergoes secondary dissociation to form C$_2$F$_4$ with a time constant of 292 ps, consistent with the previous ps-TRXL results. A comprehensive overview of population dynamics is presented in Fig. 4.

## Implications of reconstructing rotational isomerization from structural perspectives

In this work, we designed an experiment using an ultrashort laser pulse to transiently generate chemical environments far from equilibrium. We applied fs-TRXL on an illustrative model system of C$_2$F$_4$I$_2$ to elucidate the C–C and C–C• single-bond rotation dynamics coupled with rotational isomerization between multiple conformers. Using a UV pump laser pulse, the non-equilibrium states of C$_2$F$_4$I$_2$ and C$_2$F$_4$I• were prepared to allow several unique observations on their journey back to dynamic equilibria. For example, the rotational isomerization among the *non-photoreacted* molecules could be captured in this work by perturbing the mixture of C$_2$F$_4$I$_2$, in sharp contrast to the typical pump-probe experiments where only the dynamics starting from the *photoexcited* molecules could be investigated. Moreover, the non-equilibrium mixture of C$_2$F$_4$I• features the single-bond rotation along the C–C• bond with the exceptionally faster time constant (1.2 ps) than typical values in single-bond rotation kinetics. Our analysis involved a systematic extraction of major components from the TRXL data, followed by a numerical comparison with various simulated models to account for the structural origins of the dissociation and rotational dynamics. We identified how the population and structure of the conformers, as well as their relative fractions governing each reaction step, change over time. These experimental findings are also well supported by quantum calculations performed at various levels. As a final remark, we note that traditional equilibrium assumptions often do not hold for chemical and biological processes occurring far from equilibrium. Nevertheless, direct observations of such phenomena, particularly concerning rotational coherences, remain in their infancy. In this regard, the insights from this study will pave a way for a deeper comprehension of ultrafast dynamics related to internal bond rotation, such as the coherent motions associated with re-equilibration from states far from dynamic equilibrium.

## Methods
### Time-resolved X-ray liquidography at PAL-XFEL
We prepared a 60 mM solution of 1,2-tetrafluorodiiodoethane (C$_2$F$_4$I$_2$; 97%, SynQuest) in cyclohexane (C$_6$H$_{12}$; Aldrich, 99.9%), and used it for a femtosecond time-resolved X-ray liquidography (fs-TRXL) measurement. We also conducted a separate fs-TRXL measurement with a 6.4 mM solution of 4-bromo-4′-(*N,N*-diethylamino)-azobenzene (HANCHEM, 99.9%), also referred to as Azo-Br, dissolved in cyclohexane (C$_6$H$_{12}$; Aldrich, 99.9%) to elucidate the effect of the response of bulk solvent (see section 2-3 of SI). The fs-TRXL experiments were carried out at the XSS-FXL beamline of PAL-XFEL in Korea.

In the experiment, the C$_2$F$_4$I$_2$ solution was excited with a femtosecond laser pulse at 267 nm to initiate the photodynamics. After a selected time delay, the X-ray scattering images, which encrypt the structural dynamics of the sample, were generated by the femtosecond X-ray pulses from the XFEL. The 267 nm optical laser pulses, generated as the third-harmonics of a Ti:Sapphire laser, were focused

**Table 1 | Experimental and theoretical kinetic constants assigned to structural dynamics**

|  | $\tau_1$ (fs) | $\tau_2$ (ps) | $\tau_3$ (ps) | $\tau_4$ (ps) |
|---|---|---|---|---|
| Experiment | $130 \pm 50$ | $1.2 \pm 0.4$ | $26 \pm 2.4$ | $292 \pm 136$ |
| Theory | 40[a] | 1.15[b] | 20.2[b] |  |
| Dynamics | primary iodine dissociation | *anti*-to-*gauche* rotation of $C_2F_4I\cdot$ | *gauche*-to-*anti* rotation of $C_2F_4I_2$ | secondary iodine dissociation |

[a]The time constants were referred to from the previous study which reported the primary iodine dissociation trajectory of gas-phase $C_2F_4I_2$ using the surface hopping simulation[33].
[b]The time constants were computed through the Rice-Ramsperger-Kassel-Marcus (RRKM) theory, as detailed in the main text.

**Table 2 | Optimized structural parameters of the *anti* and *gauche* conformers of the two reacting species, $C_2F_4I_2$ and $C_2F_4I\cdot$**

|  | *anti*-$C_2F_4I_2$ | | | *gauche*-$C_2F_4I_2$ | | | |
|---|---|---|---|---|---|---|---|
|  | $r_{CI}^A$ (Å) | $\theta_{CCI}^A$ (°) | $\theta_{pl}^A$ (°) | $r_{CI}^G$ (Å) | $\theta_{CCI}^G$ (°) | $\theta_{pl}^G$ (°) | $\varphi_{dihed}^G$ (°) |
| Theory[a] | 2.20 | 115 | 121 | 2.20 | 116 | 120 | 66 |
| ps-TRXL[b] | $2.15 \pm 0.03$ | $112 \pm 9.8$ | 124[c] | $2.14 \pm 0.05$ | $116 \pm 10.4$ | 121[c] | 66[c] |
| fs-TRXL | $2.12 \pm 0.007$ | $110 \pm 0.66$ | $120 \pm 0.14$ | $2.10 \pm 0.034$ | $114 \pm 0.26$ | $122 \pm 0.25$ | $66 \pm 0.97$ |
|  | *anti*-$C_2F_4I\cdot$ | | | *gauche*-$C_2F_4I\cdot$ | | | |
|  | $r_{CI}^a$ (Å) | $\theta_{CCI}^a$ (°) | $\theta_{pl}^a$ (°) | $r_{CI}^g$ (Å) | $\theta_{CCI}^g$ (°) | $\theta_{pl}^g$ (°) | $\varphi_{dihed}^g$ (°) |
| Theory[a] | 2.22 | 115 | 137 | 2.20 | 113 | 137 | 66 |
| ps-TRXL[b] | $2.16 \pm 0.04$ | $119 \pm 8.2$ | 140[c] | $2.14 \pm 0.05$ | $118 \pm 9.4$ | 138[c] | 66[c] |
| fs-TRXL | $2.16 \pm 0.013$ | $119 \pm 1.3$ | $142 \pm 0.70$ | $2.14 \pm 0.011$ | $118 \pm 9.0$ | $139 \pm 3.8$ | $67 \pm 1.5$ |

[a]The structural parameters were obtained through the geometry optimization process using extended multi-state complete active space second-order perturbation theory (XMS-CASPT2)[49] calculations.
[b]The structural parameters were extracted from global fitting analysis of the ps-TRXL data about the photodynamics of $C_2F_4I_2$ in cyclohexane after 100 ps[37].
[c]These variables in the previous study were not used as fitting parameters but were instead fixed to the values obtained from the DFT calculations.
Structural parameters obtained from the femtosecond TRXL study in this work (denoted as "fs-TRXL") are compared with those derived from computations (denoted as "Theory") and the previous picosecond TRXL study (denoted as "ps-TRXL")[37]. The parameters investigated include the C–I distance ($r_{CI}$), the CCI angle ($\theta_{CCI}$), the plane angle between the C–I vector and FCF plane ($\theta_{pl}$), and the dihedral angle ($\varphi_{dihed}$). These parameters were characterized for four species, with superscripts (A, G, a, g) used to distinguish between *anti*-$C_2F_4I_2$, *gauche*-$C_2F_4I_2$, *anti*-$C_2F_4I\cdot$, and *gauche*-$C_2F_4I\cdot$, respectively. Overall, the structural parameters determined from this fs-TRXL study closely align with the ps-TRXL results, particularly for radicals, where the deviations are observed between the DFT values and the ps-TRXL results.

**Table 3 | Concentration-related parameters of the photo-dissociation and rotational isomerization dynamics**

|  | $f_{GS}$ | $f_{Rad}$ | $f_{exc}$ | $f_{rem}$ |
|---|---|---|---|---|
| ps-TRXL[a] | $0.84 \pm 0.037$ |  |  |  |
| fs-TRXL | $0.79 \pm 0.048$ | $0.71 \pm 0.089$ | $0.91 \pm 0.11$[b] | $0.78 \pm 0.004$[b] |
|  | $\chi_a$ | $\chi_g$ | $\chi_{exc}$ |  |
| ps-TRXL[a] | $0.052 \pm 0.0014$[b] |  |  |  |
| fs-TRXL | $0.049 \pm 0.0031$ | $0.019 \pm 0.0045$ | $0.043 \pm 0.0036$[b] |  |

[a]The parameters were extracted from global fitting analysis of the ps-TRXL data about the photodynamics of $C_2F_4I_2$ in cyclohexane after 100 ps[37].
[b]The error values for the dependent parameters ($f_{exc}$, $f_{rem}$, and $\chi_{exc}$) were calculated from the statistical error values for the independent parameters ($f_{GS}$, $f_{Rad}$, $\chi_a$, and $\chi_g$) using the error propagation formula.
The symbols in the first row represent the following: the population fraction of *anti*-$C_2F_4I_2$ in the total amount of $C_2F_4I_2$ at equilibrium ($f_{GS}$), the population fraction of *anti*-$C_2F_4I\cdot$ in the total amount of $C_2F_4I\cdot$ at equilibrium ($f_{Rad}$), the fraction of *anti*-$C_2F_4I_2$ in the dissociating population of $C_2F_4I_2$ ($f_{exc}$), the fraction of *anti*-$C_2F_4I_2$ in the remaining population of $C_2F_4I_2$ after dissociation ($f_{rem}$), the population fraction of *anti*-$C_2F_4I_2$ initially dissociated to *anti*-$C_2F_4I\cdot$ ($\chi_a$), the population fraction of *gauche*-$C_2F_4I_2$ initially dissociated to *gauche*-$C_2F_4I\cdot$ ($\chi_g$), and the total dissociation ratio from $C_2F_4I_2$ to $C_2F_4I\cdot$ ($\chi_{exc}$). The results from the previous ps-TRXL study are also presented for comparison. We note that, as this ps-TRXL study assumed that $\chi_a$ and $\chi_g$ are identical, $f_{GS}$, $f_{Rad}$, $f_{exc}$, and $f_{rem}$ were all regarded to be equal in the first row.

to a $132 \times 132\ \mu m^2$ spot size (full-width-at-half-maximum, FWHM) with a fluence of $2.20\ mJ/mm^2$ at the sample position. The temporal width of the optical laser pulse was 130 fs (FWHM). The X-ray pulses had a narrow bandwidth ($\Delta E/E \simeq 0.15\%$) centered at 12.7 keV and were delivered at a repetition rate of 30 Hz. These X-ray pulses were focused to a spot size of $23 \times 19\ \mu m^2$ (FWHM) at the sample position. The temporal width of the X-ray pulse was less than 50 fs (FWHM). The optical pump and X-ray probe were spatiotemporally overlapped at the stable position of the liquid jet, with a crossing angle of 10°. The temporal widths of optical laser and X-ray pulses, the velocity mismatch caused by the width of the capillary jet, and the timing jitter altogether decide the temporal resolution of the experiment, expressed by instrument response function (IRF). We determined the IRF value of $172 \pm 47$ fs from the data analysis.

To ensure a continuous supply of fresh molecules at the interacting point, the sample solution was circulated through a 100-μm-thick capillary nozzle at a linear velocity faster than 3 m/s. Also, the sample solution was refreshed every two to three hours to maintain the integrity of the sample and to exclude the effects of irreversible products and radiation damages during the experiment. The X-ray scattering images were collected on the Rayonix MX225-HS area detector located 39 mm downstream of the circulating jet position. We accurately calibrated the X-ray beam center position and sample-to-detector distance by measuring the X-ray diffraction pattern of the reference sample, lanthanum hexabromide ($LaB_6$), and comparing this pattern with the known diffraction peak positions.

The TRXL images were collected at various time delays ranging from −0.3 ps to 3 ps with a linear step of 50 fs, plus an additional 31 log-scaled points per decade up to 100 ps, resulting in a total of 98 time points. To maintain the temporal stability between the laser and X-ray pulses, we employed a timing feedback tool in the PAL-XFEL. To improve the signal-to-noise ratio of the data, a number of images (~8500 images) were repeatedly collected and averaged at each time delay. A typical TRXL experimental setup is depicted in Supplementary Fig. 1. Further details on the TRXL experiment can be found in previous publications[26–30,37,46].

**Reducing the data to $\Delta S_0(q, t)$**
The real space vectors, referenced at the X-ray beam center on the detector plane, were converted into reciprocal space vectors, whose

magnitudes are defined by $q = (4\pi/\lambda)\sin(\theta)$, where $\lambda$ and $2\theta$ denote the X-ray wavelength and scattering angle, respectively. We performed a series of corrections on the measured scattering images to address four crucial factors: (1) the solid angle of each detector pixel, (2) the polarization direction of the X-rays, (3) the alignment of the detector plane, and (4) attenuation arising from the phosphor screen of the detector. We used pyFAI[47], a well-known python library for X-ray scattering and diffraction image processing, to perform these corrections. These corrected images were converted into the static 1D isotropic scattering curves ($S_0$) by removing the anisotropic component ($S_2$). Afterwards, we normalized the static $S_0$ curves by dividing them with the sum of scattering intensities within the $q$ range of $3-6.5\,\text{Å}^{-1}$. These normalized $S_0$ curves were further scaled to properly represent the absolute X-ray scattering signal from one solute molecule. The $S_0$ curves taken without the laser ($S_0(q, \text{off})$) were subtracted from those taken at a time delay $t$ ($S_0(q, t)$) to generate the difference X-ray scattering data ($\Delta S_0^{\text{raw}}(q, t)$) encrypting the structural dynamics information. Details of the initial data processing scheme are provided in section 1 of SI.

### Determining the number of intermediates and kinetic constants
We decomposed $\Delta S_0^{\text{raw}}(q, t)$ into the solute-related components, $\Delta S_0(q, t)$, and the heat-related components, $\Delta S_0^{\text{heat}}(q, t)$, by implementing the projection to extract the perpendicular component (PEPC) method. The solvent heating and artifact signals are represented with $\mathbf{H}(q)$, a matrix obtained from the TRXL data of Azo-Br. The retrieval of $\mathbf{H}(q)$ and the assessment of the temperature rise of the solution are detailed in section 1-3 of SI. $\Delta S_0^{\text{raw}}(q, t)$ and $\Delta S_0^{\text{heat}}(q, t)$ are demonstrated in Supplementary Figs. 3–4.

To determine the number of intermediates and kinetic constants, we employed singular value decomposition (SVD), an algebraic technique that decomposes the data into a product of left singular vectors (LSV($q$), time-independent components), right singular vectors (RSV($t$), temporal evolutions) singular values ($S$, relative contribution of each rank). The SVD analysis of $\Delta S_0(q, t)$ revealed five components contributing to $\Delta S_0(q, t)$ (Supplementary Fig. 5). These five RSVs were globally fitted by a sum of multiexponential functions convoluted by a Gaussian-shaped IRF with $w = 172 \pm 47$ fs. Satisfactory fit results were obtained using four Gaussian-convoluted exponential functions, each characterized by the shared time constants of $130 \pm 50$ fs, $1.2 \pm 0.39$ ps, $26 \pm 2.4$ ps, and 292 ps. The discrepancy between experimental and fitted curves in the sub-ps time domain of the second RSV can be improved by replacing $130 \pm 50$ fs with two exponential constants, $105 \pm 13$ fs and $114 \pm 13$ fs. Nevertheless, we used four exponentials for the kinetic analysis, as rationalized in the main text (see section 1-5 of SI for details).

### Analysis of decay-associated difference scattering curves (DADSs)
To analyze the detailed structural origins of the extracted time constants, we retrieved the experimental DADSs (denoted as $\mathbf{DADS}(q)$) from $\Delta S_0(q, t)$ using the following equation.

$$\Delta S_0(q, t) = \mathbf{DADS}(q) \times \mathbf{C}'_{\mathbf{DADS}}(t)^{\mathrm{T}} = \sum_{k=1}^{4} \text{DADS}_k(q) \times \text{C}'_{\text{DADS}_k}(t)^{\mathrm{T}} \quad (1)$$

Here, $\text{C}_{\text{DADS}k}'(t)$ is a hypothetical kinetic profile corresponding to the process associated with $\text{DADS}_k(q)$. The superscript T indicates the transpose of a matrix. Using Eq. 1, we retrieved $\mathbf{DADS}(q)$ that minimizes the $\chi^2$-distance between $\Delta S_0(q, t)$ and $\mathbf{DADS}(q) \times \mathbf{C}_{\mathbf{DADS}}(t)^{\mathrm{T}}$. The details are provided in section 3 of SI. We extracted and analyzed $\text{DADS}_k(q)$s for a total of four time constants (130 fs, 1.2 ps, 26 ps, and 292 ps). $\text{C}_{\text{DADS}k}'(t)$ arises from zero to unity within IRF and decaying again to zero with a time constant of $\tau_k$. It is expressed by the

convolution of an exponential decay with a normalized Gaussian function centered at zero, $\mathscr{N}(w, 0; t)$, as in the following equations.

$$\text{C}'_{\text{DADS}_k}(t) = \mathscr{N}(w, 0; t) \otimes \left(\exp\left(-\frac{t}{\tau_k}\right)\right) \quad (2)$$

$$\mathscr{N}(w, 0; t) = \frac{1}{\sqrt{2\pi}w} \exp\left(-\frac{t^2}{2w^2}\right) \quad (3)$$

Here, $\otimes$ indicates convolution, and an IRF of $w = 172$ fs was used to construct $\mathscr{N}(w, 0; t)$.

After retrieving $\mathbf{DADS}(q)$, we simulated decay-associated difference scattering curves, $\mathbf{DADS}'(q)$, corresponding to each potential candidate pathway in Supplementary Fig. 6. The resulting set of candidate $\mathbf{DADS}'$ ($q$) were compared with each $\text{DADS}_k(q)$. By identifying the pathway—a pair of chemical species for initial and final states—whose difference scattering curve best matches with each $\text{DADS}_k(q)$, we can attribute the dynamic pathway to each kinetic constant ($\tau_k$). We also conducted the structure refinement when constructing each $\text{DADS}_k'(q)$ as detailed in the next section. We examined the candidate pathways in Supplementary Fig. 6 for $\text{DADS}_1(q)$ (Supplementary Fig. 8), $\text{DADS}_2(q)$ (Supplementary Fig. 9), and $\text{DADS}_3(q)$ (Supplementary Fig. 10), and attributed them to the primary I• dissociation, the *anti*-to-*gauche* rotational isomerization of $C_2F_4$I•, and the *gauche*-to-*anti* rotational isomerization of $C_2F_4I_2$, respectively. For $\text{DADS}_4$, we simulated the scattering curve using the known species and concentrations at 100 ps, and subtracted the scattering curve of the photodissociated mixture of $C_2F_4I_2$ (Fig. 3). Details of the DADS analysis is provided in section 3 of SI.

### Structure refinement
While analyzing the DADS to identify the structural transitions corresponding to each DADS, we simultaneously determined the molecular geometry of the species involved in the reaction through structure refinement. We started with the molecular geometries of the species optimized by DFT calculations and iteratively varied their structural parameters so that the simulated difference scattering curves, $\mathbf{DADS}'$ ($q$), generated from the refined structures best matched the experimental counterpart, $\mathbf{DADS}(q)$. The parameters, $r_{\text{CI}}$, $\theta_{\text{CCI}}$, $\theta_{\text{pI}}$, and $\varphi_{\text{dihed}}$, defined in Fig. 3d (with more details provided in Supplementary Fig. 7) were varied near the DFT-optimized values (Table 2). In $C_2F_4I_2$, the two carbon atoms share the four structural parameters to represent symmetry. On the other hand, in $C_2F_4$I•, the structural parameters at the $-CF_2$I site were taken to be equivalent to those for $C_2F_4I_2$ of the respective conformer (denoted with the superscripts "A" or "G"), whereas those at the $-CF_2$• site were defined with new parameters (denoted with the superscripts "a" or "g").

We utilized the reduced chi-square for the $k^{\text{th}}$ DADS, denoted as $\chi^2_{\text{v}k}$, as a statistical criterion to decide the best-matching candidate of $\text{DADS}_k'(q)$ with $\text{DADS}_k(q)$ for k = 1–3 (see section 3-1 of SI for details of the statistical criteria). $\mathbf{DADS}'(q)$ were calculated using well-established formulations from scattering theory, such as the Debye equation (see section 2-1 of SI for details). The simulated $\mathbf{DADS}'(q)$ were converted to the $r$-space, $\mathbf{DADS}'(r)$, through the Fourier sine transform (in eqs. S7–S8). We adjusted the amplitude of $\mathbf{DADS}'(q)$ to be on the exact scale by multiplying the scattering signal from one solute molecule by the relative molar quantity corresponding to each transition. As the volume of the solution at the interaction point remains identical for every species, we simply used concentrations ($c_{\text{DADS}k}$) to express relative molar quantities. The scale factors were determined from the kinetic model. Here, the first two equations were used to construct the scale factors, and the remaining four were

introduced to build an expression for each of the four $DADS_k'(q)$s.

$$c_{DADS1} = c_{DADS2} = c_{DADS4} = c_0 \chi_{exc} \tag{4}$$

$$c_{DADS3} = c_0 (1 - \chi_{exc}) \tag{5}$$

$$DADS_1'(q) = c_{DADS1} \cdot \begin{Bmatrix} (f_{exc} \cdot S_A(q) + (1 - f_{exc}) \cdot S_G(q)) \\ -(f_{exc} \cdot (S_a(q) + S_I(q)) + (1 - f_{exc}) \cdot (S_g(q) + S_I(q))) \end{Bmatrix} \tag{6}$$

$$DADS_2'(q) = c_{DADS2} \cdot \left\{ (f_{Rad} - f_{exc}) \cdot (S_a(q) - S_g(q)) \right\} \tag{7}$$

$$DADS_3'(q) = c_{DADS3} \cdot \left\{ (f_{rem} - f_{GS}) \cdot (S_G(q) - S_A(q)) \right\} \tag{8}$$

$$DADS_4'(q) = c_{DADS4} \cdot \begin{Bmatrix} 0.7 \cdot (f_{Rad} \cdot (S_a(q) + S_I(q)) + (1 - f_{Rad}) \cdot (S_g(q) + S_I(q))) \\ +0.3 \cdot (S_P(q) + 2 S_I(q)) - (f_{exc} \cdot S_A(q) + (1 - f_{exc}) \cdot S_G(q)) \end{Bmatrix} \tag{9}$$

Here, $f_{GS}$ is the population fraction of *anti*-$C_2F_4I_2$ in the total amount of $C_2F_4I_2$ at equilibrium, $f_{Rad}$ is the population fraction of *anti*-$C_2F_4I\bullet$ in the total amount of $C_2F_4I\bullet$ at equilibrium, $\chi_a$ is the population fraction of *anti*-$C_2F_4I_2$ initially dissociated to *anti*-$C_2F_4I\bullet$, $\chi_g$ is the population fraction of *gauche*-$C_2F_4I_2$ initially dissociated to *gauche*-$C_2F_4I\bullet$, $\chi_{exc}$ is the total dissociation ratio from $C_2F_4I_2$ to $C_2F_4I\bullet$, $f_{exc}$ is the fraction of *anti*-$C_2F_4I_2$ in the dissociating population of $C_2F_4I_2$, and $f_{rem}$ is the fraction of *anti*-$C_2F_4I_2$ in the remaining population of $C_2F_4I_2$ after dissociation. $c_0$ indicates the initial $C_2F_4I_2$ concentration ( = 60 mM). $S_{subscript}(q)$ represents the simulated X-ray scattering curve from the species denoted with the subscript, where the subscripts (A, G, a, g, P, and I•) stand for the abbreviation of *anti*-$C_2F_4I_2$, *gauche*-$C_2F_4I_2$, *anti*-$C_2F_4I\bullet$, *gauche*-$C_2F_4I\bullet$, $C_2F_4$, and I•, respectively. Among the concentration-related parameters, the first four ($f_{GS}$, $f_{Rad}$, $\chi_a$, and $\chi_g$) were iteratively optimized, while the later three ($\chi_{exc}$, $f_{exc}$, and $f_{rem}$) were calculated from them using the equations in section 3-3 of SI. The method to construct simulated scattering curves from the molecular structure is described in section 2-1 of SI. The theory for the DADS analysis is summarized in section 3 of SI.

## Analysis using linear combination fit (LCF)
Using the optimal model determined from the analysis of the DADSs, we constructed the simulated species-associated difference scattering curves (**SADS'**$(q)$). To obtain **SADS'**$(q)$, we took the following five species into consideration: (1) *anti*-$C_2F_4I\bullet$, (2) *gauche*-$C_2F_4I\bullet$, (3) *anti*-$C_2F_4I_2$, (4) *gauche*-$C_2F_4I_2$, and (5) $C_2F_4$. Then, SADS'$_k(q)$s were calculated by subtracting the scattering curve for the initially reacting mixture of $C_2F_4I_2$ from that corresponding to the $k^{th}$ species.

$$SADS_k'(q) = S_k(q) - (f_{exc}S_A(q) + (1 - f_{exc})S_G(q)) \tag{10}$$

Here, $f_{exc}$ is the fraction of *anti*-$C_2F_4I_2$ in the dissociating population of $C_2F_4I_2$. Also, $S(q)$ indicates the simulated X-ray scattering curve from the species denoted with the subscript, where the subscripts A and G mean *anti*-$C_2F_4I_2$ and *gauche*-$C_2F_4I_2$, respectively. Initially, the ground state ensemble consists of a mixture of *anti* and *gauche* $C_2F_4I_2$. Upon photoexcitation, the portion of $C_2F_4I_2$ with an *anti*-to-*gauche* ratio of $f_{exc}$:$(1-f_{exc})$ are depleted from the original equilibrium composition and start to follow the photoreaction pathway, as described in the subtracted term of Eq. 10. Here, it is noteworthy that the SADSs for (3) and (4) have identical shapes but opposite signs, as both components are mathematically represented by the difference between *anti* and *gauche* $C_2F_4I_2$. Therefore, we evaluated the scattering contributions

from both *anti*-$C_2F_4I_2$ and *gauche*-$C_2F_4I_2$ collectively and merged the two components into one (explained as the "third" species, k = 3; therefore, the SADS for $C_2F_4$ becomes the "fourth" species, k = 4).

$$\begin{aligned} & SADS_A'(q) \times C_{SADS,A}(t)^T + SADS_G'(q) \times C_{SADS,G}(t)^T \\ & = (S_A(q) - (f_{exc}S_A(q) + (1 - f_{exc})S_G(q))) \times C_A(t)^T \\ & \quad + (S_G(q) - (f_{exc}S_A(q) + (1 - f_{exc})S_G(q))) \times C_G(t)^T \\ & = (S_A(q) - S_G(q)) \times (C_A(t) - f_{exc}(C_A(t) + C_G(t)))^T \\ & = SADS_3'(q) \times C_{SADS3}(t)^T \end{aligned} \tag{11}$$

Here, Eq. 11 has a similar form as Eq. 10 since it is expressed as a multiplication of the time-dependent concentration part and the time-independent $q$-space part.

Subsequently, we retrieved the LCF concentration matrix, **C**$_{SADS}(t)$ (see Fig. 3e), that minimizes the reduced $\chi^2$ value between the solute-related experimental data ($\Delta S_0(q, t)$) and the simulated data (**SADS'**$(q) \times$ **C**$_{SADS}(t)^T$). Details of the LCF analysis are assessed in section 4 of SI. Since the amplitude of the residual signal is negligible (see Supplementary Fig. 13), we concluded that the LCF model adequately explains the TRXL data and effectively summarizes the structural dynamics of $C_2F_4I_2$. We also provide an alternative analytical approach based on SADSs in Supplementary Fig. 12, whose details are described in section 3-3 of SI.

## Ab-initio and density functional theory (DFT) calculations
Molecular geometries of all reacting species, including *anti*-$C_2F_4I_2$, *gauche*-$C_2F_4I_2$, *anti*-$C_2F_4I\bullet$, *gauche*-$C_2F_4I\bullet$, $C_2F_4$, and $C_2F_4...I_2$ (species "X") were optimized using DFT. The ωB97X functional[48], well-known for its ability to optimize geometry in line with experimental observations[26,31,32,37,46], was utilized. These DFT-optimized structures served as the initial points for the structure refinement (see Supplementary Data 1).

To explore the nature of excited states of $C_2F_4I_2$, potential energy curves (PECs) with respect to the C−I distance were calculated for *anti* and *gauche* $C_2F_4I_2$ in extended multi-state complete active space second-order perturbation (XMS-CASPT2) theory[49], from the SA(9 singlets, 11 triplets)-CAS(12e, 10o)SCF wavefunctions. Diabatic PECs were obtained, with the semi-diabatization process from the calculated adiabatic PECs. For obtaining accurate energy barriers during the C−C bond rotation, Gibbs free energies were calculated for the two transition states, namely T1 and T2, for the four species involved in rotational isomerization (*anti*-$C_2F_4I_2$, *gauche*-$C_2F_4I_2$, *anti*-$C_2F_4I\bullet$, and *gauche*-$C_2F_4I\bullet$). Furthermore, PECs with respect to the dihedral angles (ICCI for $C_2F_4I_2$ and ICCF for $C_2F_4I\bullet$) were calculated with three different levels of theory (CCSD(T), XMS-CASPT2, and DFT (ωB97X functional)) for the four species. The results were demonstrated in Table 2. We note that (7e, 5o) was employed for the active space of $C_2F_4I\bullet$, analogous to the active space of $C_2F_4I_2$.

To calculate vertical excitation energies and PECs of $C_2F_4I_2$ with respect to the C−I distance, ANO-RCC-VDZP all-electron basis set was employed. We utilized the LANL2DZ basis set for CCSD(T) and XMS-CASPT2 calculations, and the def2-TZVPP basis set for DFT calculations. Openmolcas 8.6[50] was used for the multireference calculations, ORCA 5.0.4[51] was used for the CCSD(T) calculations, and Gaussian 16[52] was used for DFT calculations.

## Molecular dynamics simulation
Molecular dynamics (MD) simulations were conducted via the MOLDY[53] program to model the interaction between each chemical species involved in the reaction and surrounding solvent molecules. Here, we employed a virtual cubic box composed of one solute molecule surrounded by 512 rigid solvent molecules. The internal structures of each molecule (*anti*-$C_2F_4I_2$, *gauche*-$C_2F_4I_2$, *anti*-$C_2F_4I\bullet$, *gauche*-$C_2F_4I\bullet$, and I•) were fixed to the optimized ones from the DFT

method, as detailed in the "Density functional theory calculation" section. The partial charge distributions of each molecule were determined by natural population analysis (NPA) using the optimized structures. In the simulation, the molecular dynamics were governed by intermolecular interactions modeled with Coulombic forces and Lennard-Jones potentials. All simulations took place at an ambient temperature of 300 K and a solvent density of 0.779 g/cm³. We initiated system equilibration via coupling to a Nose-Hoover thermostat over a 20 ps duration. The simulations were conducted within the NVT ensemble, utilizing a time step of 100 as, and trajectories were traced for a duration of 1 ns. During the simulation, molecular snapshots were saved at intervals of 100 ps. From these molecular snapshots, we first extracted the pair distribution functions (PDFs) and calculated the theoretical X-ray scattering curves for the cage term, $S_{cage}$. This process is described in section 2-1 of SI.

### Simulating the rotational isomerization dynamics

We used the Rice-Ramsperger-Kassel-Marcus (RRKM) theory[43–45] to calculate the theoretical kinetic constants for the rotational isomerization of $C_2F_4I_2$ and $C_2F_4I\bullet$. We assumed that the rotational isomerization dynamics of both $C_2F_4I_2$ and $C_2F_4I\bullet$ takes a path trespassing either of the two transition states, T1 and T2. We utilized the optimized molecular structures, vibrational frequencies, electronic energy levels, and free energies for *anti*, *gauche*, T1, and T2 states of $C_2F_4I_2$ and $C_2F_4I\bullet$. The dynamic equilibrium of the forward (*anti*-to-T2-to-*gauche*) and the reverse (*gauche*-to-T2-to-*anti*) pathways leads to identical difference scattering signals with opposite signs, quenching each other in the kinetic picture. Therefore, only the dynamics arising from the re-equilibration of the perturbed state to the dynamic equilibrium are observed in our experiment. The potential energy surface with respect to the dihedral angle implies that the rotational isomerization takes place through the T2 transition state. In this context, only the theoretical kinetic constants for the assigned rotational dynamics (*gauche*-to-*anti* for $C_2F_4I_2$, *anti*-to-*gauche* for $C_2F_4I\bullet$) passing through the corresponding T2 states were compared with experimentally extracted values in Table 3. All computations based on the RRKM theory were conducted using the *ChemRate* software[54–56].

In the RRKM calculation, the rate constant is expressed by the following equation.

$$k = a_1 \times \exp\left(-\frac{a_2}{RT}\right) \quad (12)$$

Here, the gas constant R is 8.314 J·mol⁻¹·K⁻¹, the temperature T was assumed to be 300 K, and $a_1$ and $a_2$ were computed through the RRKM theory. From the computation results, we found that $a_1 = 9.48 \times 10^9$ and $a_2 = -4117.71$ in the *gauche*-to-T2-to-*anti* pathway of $C_2F_4I_2$, and $a_1 = 4.56 \times 10^{11}$ and $a_2 = -1613.45$ in the *anti*-to-T2-to-*gauche* pathway of $C_2F_4I\bullet$. At 300 K, these values lead to the time constants of 20.2 ps for the $C_2F_4I_2$ rotation and 1.15 ps for the $C_2F_4I\bullet$ rotation, as discussed in the main text.

### Reporting summary

Further information on research design is available in the Nature Portfolio Reporting Summary linked to this article.

## Data availability

Source data for the figures in the main text and Supplementary Figs. are provided with this paper. The uploaded source data includes the processed data that can reproduce the entire key findings of this study. The raw TRXL data, which is not uploaded due to capacity issues, is securely stored on the archive servers of the beamline facilities and will be made available upon requests from readers. All other data generated in this study to support the findings of this study will also be available from the authors upon request. Source data are provided with this paper.

## Code availability

The code utilized for the analysis of time-resolved X-ray liquidography data can be obtained from the corresponding author upon request.

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

## Acknowledgements

This work was supported by the Institute for Basic Science (IBS-R033) granted to H.I. The experiments were performed using the XSS-FXL beamline at PAL-XFEL (Proposal No. 2020-2nd-XSS-001) funded by the Ministry of Science and ICT of Korea. We acknowledge M. Kim, H. Hyun, J. K. Park. S. Kim, S.-Y. Park, and other scientists in Pohang Accelerator Laboratory X-ray Free Electron Laser (PAL-XFEL) for the supports during the beamtime. We also thank J. G. Kim for experimental aids and fruitful discussions.

## Author contributions

Conceptualization, S.L. and H.I.; Methodology, S.L., H.K., D.I., A.S., and H.I.; Validation, S.L.; Formal analysis, S.L.; Investigation, S.L., H.K., Ju.K., Y.L., J.G., J.H., Y.C., K.W.L., D.K., Je.K., R.M., J.H.L., and H.I.; Resources, S.L., R.M., and J.H.L.; Data Curation, S.L. and J.H.L.; Writing – Original draft, S.L., H.K. D.I., and H.I.; Writing – Review & Editing, S.L., H.K., D.I., Je.K., and H.I.; Visualization, S.L., H.K., D.I., and H.I.; Supervision, H.I.; Project Administration, H.I.; Funding Acquisition, H.I.

## Competing interests

The authors declare no competing interests.
