## [Transparent Peer Review file · Nature Communications]

Ultrafast structural dynamics of carbon–carbon single-bond rotation in transient radical species at non-equilibrium

Corresponding Author: Professor Hyotcherl Ihee

Version 0:

Reviewer comments:

Reviewer #1

(Remarks to the Author)

The paper “Ultrafast structural dynamics of carbon-carbon single-bond rotation in transient radical species at non-equilibrium” elucidates on the structural dynamics of ethane derivatives, which were modified with iodine and fluorine substitutes, to investigate molecular rotations of C-C single bonds. The substitution with heavy atoms allows the use of time-resolved X-ray scattering for good structural sensitivity, which revealed a time scale of 1.2 ps for the C-C bond rotation of excited C2F4I* from anti isomers towards gauche isomers.

The study gains impact by simultaneously observing excited state C2F4I* anti isomers as well as non-excited C2F4I2 gauche isomers, which are found to rotate around the single bond within 26 ps. Both time scales appear to be relevant for a broader range of ethane derivatives.

The analysis is a tour de force of advanced and state of the art methods, and the conclusions of the paper seem to be well supported by experimental and simulated data.

However, some minor points should be addressed:

Abstract: The reason for using substituted ethane and the fact that C2F4I* and C2F4I2 were measured simultaneously in the experiment are not clear from the abstract. The explanation in the introduction is well-written, but the abstract would benefit from a clearer picture of the experiment conducted.

Artefacts are apparent in Figure S3, especially in S3a between 1 and 10 ps and in S3b in the positive feature around $q=1.9 \text{ \AA}^{-1}$. This is somewhat worrying, and the authors must more clearly explain how this data quality, from which the solvent response was determined, might impact the precision of the data analysis of the C2F4I2 and C2F4I* compounds.

Regarding the results presented in Table 1: Please comment on how populations in the process of rotation are taken into account, considering that they cannot be considered purely anti or gauche isomers? Additionally, please comment on the use of statistical error values used for estimating the errors, as they seem very small and justify their use considering the likely presence of artifacts and/or systematic errors in the data.

Related, P.6 ll. 163 mentions ‘statistical criteria’ for the identification of DADSs. Please reference the methods used here for better clarity. Also, please justify their use, for example considering the artifacts shown in Figure S3.

P.10 line 279 states that ‘typical’ pump-probe experiments only determine dynamics of photoexcited molecules. While the observation of non-photoreacted molecules is indeed a strength of this article, a reference to other ‘untypical’ pump-probe experiments observing solvation cage dynamics or ground state dynamics could better contextualize the results of this study.

In Figure 4a, the rotation of C2F4I* from anti to gauche is represented as occurring through the T1 state with a height of 3.4 kcal/mol. How can this process happen in 1.2 ps, while the transition through the 4.0 kcal/mol T1 state of the gauche C2F4I2 molecule towards the anti state takes 26 ps?

P.24 in the main text and p.18 in the SI seem to contradict each other. Please clarify if the rotational isomerization takes

place through the T1 or the T2 transition state. Are the experimentally extracted values shown in Table 1 or Table 2?

Detailed comments:

The last row in Table 1 is split into two lines, separating the error bars into two parts. Please correct the formatting.

P.6 II. 137-138, From context, the meaning can probably be constructed, but the wording makes it unclear: "... the time-independent fingerprints of each structural dynamics and their temporal evolutions, respectively". I.e.: If the fingerprints are time-independent, how do they show a time evolution? Clearer wording might help the reader.

In the graphical abstract figure, the red vertical line represents the photoexcitation. This makes sense in the top part of the figure. However, the current version implies that anti-C₂F₄I* and gauche-C₂F₄I₂ get excited again before starting to rotate (in the center and bottom boxes). Please clarify this representation.

The shading of the 'unknown' region in Figure 1 reduces the clarity and interpretability of the figure. Additionally, the figure caption describes the gauche conformers as "yellow" in line 303, while the color in a printed as well as the online version of the manuscript is green.

The data in Figure 2 appears very crowded at early time points, impeding the identification of the time constant at 1.2 ps. Please consider revising the figure to improve clarity.

P.8, the referenced Figure S15 does not exist.

P.4, the numbers 34 and 35 in the citation are not separated by comma.

Reviewer #2

(Remarks to the Author)

Reviewer #3

(Remarks to the Author)

The internal bond rotation in molecules plays a crucial role in determining reactivity, stereochemistry, and the outcomes of reactions. As a result, the associated dynamics have been extensively studied both theoretically and experimentally. However, an atomic-level understanding of the structural dynamics of ultrafast single-bond rotation has remained elusive, primarily due to the challenge of achieving both an ultrafast timescale and the molecular structure resolution required for such studies. This manuscript addresses this challenge by reporting recent advances in directly probing the photo-induced ultrafast structural dynamics of carbon-carbon single-bond rotation in transient species of photoexcited 1,2-tetrafluorodiiodoethane (C₂F₄I₂) using femtosecond time-resolved X-ray liquidography (TRXL).

In the experiment, a highly non-equilibrium state was induced by an fs UV pump pulse, resulting in a distribution of anti- and gauche-conformers of the tetrafluorodiiodoethyl radical (C₂F₄I•) and 1,2-tetrafluorodiiodoethane. The population evolution of these transient species, as they relax back to thermal equilibrium, was monitored in real time using TRXL, which allowed the conformer ratios and structures of each reacting species to be captured. Combined with extensive theoretical simulations, the study reveals that the rotational isomerization of C₂F₄I• and C₂F₄I₂ follows anti-to-gauche and gauche-to-anti pathways with time constants of 1.2 ps and 26 ps, respectively. For the first time, this work provides atomic-level insight into the kinetics and structural dynamics of single-bond rotation.

Overall, this is a cutting-edge, solid, and thorough study, suitable for publication in a high-profile journal. However, I have a few points and questions that should be addressed before final acceptance:

(1) In lines 231 to 232 on page 9, should "the depletions of interatomic distances in anti-C₂F₄I₂ and the formations in gauche-C₂F₄I₂" be revised to "the depletion of gauche and the formation of anti-C₂F₄I₂" instead?

(2) The current experiments were conducted under a 267-nm pump light. How would the branching ratios, rotation time constants, and structures of the transient species change with varying pump photon energy? Additionally, how might the rotation dynamics differ in gas and liquid phases? In my opinion, addressing these questions would provide a broader perspective and further strengthen the paper.

Reviewer #4

(Remarks to the Author)

This is a very interesting manuscript on the ultrafast photoinduced reactions of 1,2-tetrafluorodiiodoethane (C₂F₄I₂) in solution as visualized with femtosecond solution X-ray scattering. The studied molecule features a C-C single bond that allows "the two ends" to rotate around the connecting C-C bond and the question posed here was whether and in which way

photoexcitation of the molecule would make this C-C bond rotate, either in the ensemble of intact (parent) molecules or in the radical photoproduct C2F4I after I dissociation. Seeing such rotational isomerization or bond rotation is not easy because they are often masked by other, more drastic changes of structures (such as dissociation) and it is not clear a priori how one can be sensitive to such bond rotations anyway. It is evident, though, that it is essential to probe or visualize and understand bond rotations such as those of C-C bonds as they evidently influence the course of important chemical reactions and biological processes. Any step that brings us closer to visualizing C-C bond rotations in real time to then understand the underlying rotational isomerization reactions is highly impactful in a broad range of areas, I think. The authors found a great trick, I find, to do such a step and to visualize in real time photoinduced C-C bond rotations of a molecule in solution by placing heavy F and I atoms in their molecule and seeing how these move with respect to each other with time-resolved X-ray scattering. This enabled them to infer the underlying C-C bond rotation dynamics.

The presented investigation builds on an earlier study by part of the same authors with the same method, this time with measurement conducted at a femtosecond X-ray free-electron laser source where the former study was performed at a synchrotron radiation source. Clearly, the improvement in temporal resolution from (order of magnitude) 100 picoseconds (ps) to 100 femtoseconds (fs) enabled the authors of the present investigation to reveal new and very interesting information. This study is also a great illustration of the capabilities of X-ray lasers and the new opportunities for studies of condensed-phase chemical reactions arising there.

The manuscript is very well written with very clear figures and a clear and systematic presentation of the results. The main text and supporting information complement each other very well and allow the reader to easily switch from broad to detailed views. The science is sound, and the data and analysis appear to be technically valid. Overall, the manuscript describes a nice and consistent story. The manuscript, I think, is very accessible, interesting to read and should appeal to a broad readership of scientists interested in seeing, understanding, and imagining chemical reactions or reaction steps in real time.

Of the two main results and claims of the study, the rotational isomerization between anti and gauche of C2F4I and the rotational isomerization between anti and gauche of C2F4I2, I find those of C2F4I (1.2 ps time constant in the kinetics of the reaction) very well motivated and substantiated. The ones of C2F4I2 (26 ps time constant), I think, are less clear and less validated. I understand that this latter aspect is an exciting given that these are rotational dynamics of the parent (intact non-dissociated) molecule that indirectly relate to the initial photoexcitation. I just find the evidence not very convincing, and I don't think that the authors "need" this aspect or detail of the study to have a great study or story. I thus suggest focusing the manuscript more on the C-C bond rotation of C2F4I and, in turn, weaken the claims on the C-C bond rotation in C2F4I2. What we are left with is still a wonderful achievement in my view that is worth reporting.

I have additional comments that the authors may consider in a potential revision of their manuscript:

I find that the term "radical" in the title fits to the main findings around C-C bond rotations in C2F4I and I think that that should be the focus of this manuscript. The abstract though (and the main text and conclusion sections in fact as well) does not express this focus but rather treats C2F4I and C2F4I2 seemingly equally.

In the abstract "However, structural dynamics of ultrafast single-bond rotation ... have remained elusive due to the limited sensitivity of spectroscopic tools in resolving molecular structures." is a bit misleading or not specific enough in my view. Obviously, there are numerous spectroscopic tools that are very sensitive to (changes of) molecular structures. The bond rotations addressed here, though, may be hard to detect with spectroscopic tools. I suggest reworking this sentence to better express why bond rotations are so hard to probe spectroscopically.

In the introduction in "... occurs with time constants ranging from 10 ps to 100 ps at room temperature..." it is not quite clear to me what makes this range so large (depending on what do theoretical studies predict different times?).

Version 1:

Reviewer comments:

Reviewer #1

(Remarks to the Author)

We appreciate the authors careful work in addressing the points raised in our original review report. The resubmitted work is a remarkable achievement of clear scientific interest and should be published.

Reviewer #2

(Remarks to the Author)

Reviewer #3

(Remarks to the Author)

In this revision, the authors have made adequate modifications to address the concerns raised by the reviewers in the

previous round. Overall, the revised manuscript is clearer and more cohesive. It is now ready for publication.

Reviewer #4

(Remarks to the Author)

The authors have adequately addressed and resolved my concerns. This relates to the changes made to the manuscript in the revised version, the authors' responses to the other reviewers and the response to my own questions and comments. I find that this is a very interesting and complete study, with a mature technique on an interesting problem, the real-time observation of C-C bond rotation as triggered by a short light pulse, in a great form of a well-written manuscript.

Responses to the comments from Reviewer #1

The paper “Ultrafast structural dynamics of carbon-carbon single-bond rotation in transient radical species at non-equilibrium” elucidates on the structural dynamics of ethane derivatives, which were modified with iodine and fluorine substitutes, to investigate molecular rotations of C-C single bonds. The substitution with heavy atoms allows the use of time-resolved X-ray scattering for good structural sensitivity, which revealed a time scale of 1.2 ps for the C-C bond rotation of excited C₂F₄I from anti isomers towards gauche isomers.*

The study gains impact by simultaneously observing excited state C₂F₄I anti isomers as well as non-excited C₂F₄I₂ gauche isomers, which are found to rotate around the single bond within 26 ps. Both time scales appear to be relevant for a broader range of ethane derivatives. The analysis is a tour de force of advanced and state of the art methods, and the conclusions of the paper seem to be well supported by experimental and simulated data.*

→ The authors appreciate the reviewer for a thoughtful and positive evaluation of our work.

However, some minor points should be addressed:

Abstract: The reason for using substituted ethane and the fact that C₂F₄I and C₂F₄I₂ were measured simultaneously in the experiment are not clear from the abstract. The explanation in the introduction is well-written, but the abstract would benefit from a clearer picture of the experiment conducted.*

→ We appreciate the fruitful comment provided by the reviewer. We have modified the abstract in response to the reviewer’s comment. Firstly, the reason for using substituted ethane is more detailed. We also emphasized the fact that C₂F₄I• and C₂F₄I₂ were measured simultaneously in the experiment by adding the term “simultaneously” in the later parts of the abstract. Due to the limitations of word counts in the abstract, however, we kindly ask for the reviewer’s understanding that we have no choice but to only connotatively introduce the suggested improvements. The modified abstract is as follows. We note that we highlighted the modified sentences and phrases with bold fonts.

“Bond rotation is an important phenomenon governing the fate of reactions. **In particular, heterogeneously substituted ethane derivatives provide distinct structural conformations around the bond, empowering them as ideal systems for studying the rotation along carbon-containing**

single bonds. However, structural dynamics of ultrafast single-bond rotation, especially along C–C• bonds, have remained elusive as tracking the detailed changes in structural parameters during the rotational isomerization is challenging with conventional spectroscopic tools. Here, we employed femtosecond time-resolved X-ray liquidography (TRXL) to visualize the rotational isomerization between *anti* and *gauche* conformers of tetrafluoroiodoethyl radical (C₂F₄I•) and 1,2-tetrafluorodiiodoethane (C₂F₄I₂), **simultaneously**. The TRXL data captured perturbations in conformer ratios and structures of each reacting species, revealing that the rotational isomerization of C₂F₄I• and C₂F₄I₂ follows *anti*-to-*gauche* and *gauche*-to-*anti* paths with time constants of 1.2 ps and 26 ps, respectively. These findings also align with the computational predictions. This work offers an atomic-level insight into the kinetics and structural dynamics of single-bond rotation.”

Artefacts are apparent in Figure S3, especially in S3a between 1 and 10 ps and in S3b in the positive feature around $q=1.9 \text{ \AA}^{-1}$. This is somewhat worrying, and the authors must more clearly explain how this data quality, from which the solvent response was determined, might impact the precision of the data analysis of the C₂F₄I₂ and C₂F₄I compounds.*

→ We are aware of the time-oscillating artifacts in the TRXL data of the heating dye solution ($\Delta S^H(q, t)$) in **Figure S3a**. Nevertheless, we emphasize that only the LSVs ($H(q)$) of $\Delta S^H(q, t)$ were used in retrieving the heat-related contribution (**Figure S3b**) from the data, as explicitly detailed in section 1-3 of SI. As the seemingly unstable signals along the temporal domain are reflected only in the RSVs and not in the LSVs, we ascertain that the temporal instability in $\Delta S_H(q, t)$ will not affect the overall analysis.

To further support our claim, we extracted the 1st LSV from $\Delta S^H(q, t)$ and compared it with a well-known feature of the $(\partial S/\partial T)_p$ component in cyclohexane (cited from Figure 5b of the previous study [*Phys. Chem. Chem. Phys.* **15**, 15003–15016 (2013)]) in **Figure R1**. This figure demonstrates that the $H(q)$ reproduces the known shape with far better signal-to-noise (S/N) quality despite the apparent temporal instability in $\Delta S^H(q, t)$. Furthermore, we note that the PEPCed data (**Figure S2c**) retains similar data quality to the raw isotropic data (**Figure S2b**) without introducing any traces of artifacts mentioned in the comment. This also implies that the S/N quality of $H(q)$ is sufficient to avoid creating any artifacts when orthogonalizing $\Delta S^{\text{raw}}(q, t)$ using $H(q)$, and that the credibility of further analysis will remain intact.

Comparison of cyclohexane $(dS/dT)_\rho$ terms in $q\Delta S_0^H(q, t)$

Figure R1. Comparison of cyclohexane $(\partial S/\partial T)_\rho$ terms. We superimposed the $(\partial S/\partial T)_\rho$ terms from this study (red) with the result from a previous study by K. S. Kjaer et al. (blue). The figure demonstrates that the positive feature at 1.9 \AA^{-1} referred to by the reviewer is not an experimental artifact but an actual shape of the cyclohexane $(\partial S/\partial T)_\rho$ term.

In addition, we revised the contour level in Figure S3a to make it directly comparable to that in Figure S3b (from $\pm 2,000$ to $\pm 10,000$). We believe this modification prevents giving readers a misleading impression that overly emphasizes non-significant artifacts.

Regarding the results presented in Table 1: Please comment on how populations in the process of rotation are taken into account, considering that they cannot be considered purely anti or gauche isomers?

→ As the reviewer correctly pointed out, the molecular structures of $C_2F_4I_2$ and $C_2F_4I\bullet$ deviate from those at their *anti* and *gauche* conformers during the rotational isomerization. However,

we emphasize that the TRXL data is the scattering signals not from each individual molecule but averaged over the ensemble of molecules. Therefore, during the rotational isomerization, the molecular structures that are neither purely *anti* nor purely *gauche* can only be captured experimentally if the dynamics occur coherently among the molecules within the ensemble. However, in our TRXL data, no non-exponential behaviors were observed on the timescales at which the rotational isomerizations take place (1.2 ps and 26 ps), indicating that the motions of the rotationally isomerizing molecules are out-of-phase (incoherent). In other words, the dynamics of rotational isomerization can be fully explained in terms of population dynamics, without invoking contributions from nonequilibrium structures.

Additionally, please comment on the use of statistical error values used for estimating the errors, as they seem very small and justify their use considering the likely presence of artifacts and/or systematic errors in the data.

→ First of all, the error values reported in **Extended Data Table 1** are not beyond what is generally expected from a TRXL experiment. For example, the ps-TRXL study of CeCl₆ reported the error values ranging from 0.001 Å to 0.02 Å for the Ce–Cl distances (Table 1 of [*J. Am. Chem. Soc.* **145**, 23715 (2023)]). The fs-TRXL study of I₃[−] also reported the error values of 0.01 Å for the I–I distances and 0.3° for the I–I–I angles (Figure 2b of [*Nat. Commun.* **13**, 522 (2022)]). Therefore, the error levels observed in this study (0.007 to 0.03 Å for distances and 0.14° to 3.8° for angles) lie within the boundary expected for a well-conducted TRXL experiment at XFELs or high-end synchrotrons. We note that these error values, the estimation of which is explained below, address random noise only and do not account for the potential presence of systematic errors or artifacts in the data. If such systematic errors or artifacts contribute to our data, the actual error values of the parameters may differ from the calculated values. However, since there is no evidence to suggest that systematic errors or artifacts contribute to our data, we have not made additional considerations regarding potential distortions in error values caused by such factors.

We estimate the standard error of the parameters as follows. (1) We calculated the statistical standard error of the mean of the TRXL data for each pair of q and t (defined as $\sigma(q,t)$). (2) We evaluated how the chi-square (χ^2) value changes as the values of the fitting parameters (X_1, \dots, X_n) are varied during the fitting process. (3) Based on the result from (2),

where the χ^2 values were examined as functions of the fitting parameters, we calculated the Hessian matrix $H_{ij} = \partial(\chi^2)/\partial X_i \partial X_j$ for $i, j = 1, 2, \dots, n$. (4) Assuming asymptotic normality, the standard error for each parameter is calculated as the square root of two times the diagonal elements of the inverse of the Hessian matrix. For example, the standard error of the parameter X_k is $([2 \times H^{-1}]_{kk})^{1/2}$. We added the following section to the SI of the revised manuscript.

“2-3. Estimation of standard errors

We estimated the statistical errors of the experimental data and the fitted parameters as follows: (1) estimating the standard error of the mean of the experimental TRXL curves (defined as $\sigma(q, t)$), (2) calculating how the chi-square (χ^2) value of a model varies as a function of the fitted parameters during the optimization process, (3) evaluating the Hessian matrix of χ^2 on a space spanned by the parameters, and (4) retrieving the standard error, which is the square root of two times the diagonal elements of the inverse of the Hessian matrix.

The first step aims to extract the standard error of the mean, $\sigma(q, t)$. As described in section 1-1 of SI, the experimental TRXL data used in the analysis is the average of multiple difference scattering curves, each representing the difference between two static curves (one taken after the photoexcitation and the other taken without the laser). The standard error is calculated by taking the square root to the deviation divided by the number of images at each time delay minus one. Finally, the standard error of the mean is calculated by further dividing the standard error by the square root of the number of images.

The second step focuses on the evaluation of goodness-of-fit. In this study, we used the concept of chi-square (χ^2) and reduced chi-square (χ^2_{ν}), which are textbook criteria to statistically evaluate how well the model fits to the data. Here, the χ^2 value is calculated as the mean-square distance divided by the square of $\sigma(q, t)$ summed over the entire (q, t) space. The χ^2_{ν} value is obtained by dividing χ^2 by $(n_q \times n_t - p - 1)$, where n_q is the number of q -space points, n_t is the number of simultaneously fitted curves (equals to the number of time constants in the DADS or SADS analyses, and the number of time points in the global fit analysis), and p is the number of parameters. The detailed formulas utilized specifically for this study are provided in eqs. S12 and S13. The optimization process is essentially finding the set of parameters, denoted as (X_1, X_2, \dots, X_p) , that minimizes the χ^2 value. We note that these symbolic notations for the parameters will be used only in this section (section 2-3 of SI).

The third and fourth steps focus on the evaluation of the Hessian matrix, which assesses how the goodness-of-fit becomes worse if the set of optimized parameters slightly deviates from their global

minimum. The Hessian matrix H has a size of p by p whose ij^{th} elements defined as $H_{ij} = \partial^2\chi^2/\partial(X_iX_j)$. If we apply the Taylor expansion to χ^2 near the global minimum set of (X_1, X_2, \dots, X_p) , this Hessian matrix divided by 2! will be the coefficient for the square of parameter deviation $(\Delta X_1, \Delta X_2, \dots, \Delta X_p)$. Accordingly, the standard error of the k^{th} parameter is the square root of two times the corresponding diagonal term of the inverse Hessian matrix, $([2 \times H^{-1}]_{kk})^{1/2}$. This value is reported as the error of each parameter in **Tables 1–3**.

We note that this estimation addresses random noise only and does not account for the potential presence of systematic errors or artifacts in the data. If such systematic errors or artifacts contribute to our data, the actual error values of the parameters may differ from the calculated values. In this regard, the seemingly small errors can be interpreted as precision rather than accuracy. However, as we found no evidence of systematic errors or artifacts in our data, we have not made additional considerations regarding potential distortions in error values caused by such factors.”

Related, P.6 II. 163 mentions ‘statistical criteria’ for the identification of DADSs. Please reference the methods used here for better clarity. Also, please justify their use, for example considering the artifacts shown in Figure S3.

→ For clarity, we revised the sentence containing the expression “statistical criteria” as follows.

“We identified the best candidates, which have the smallest reduced chi-square (χ_v^2) values, to describe the four DADSs (Fig. 3a). χ_v^2 is widely utilized as a statistical criterion to evaluate the fitness of models, whose details can be found in section 3-1 of SI.”

In the sentence mentioned by the reviewer, we described the following process: to determine the structural change model that best fits each DADS, we calculated theoretical curves corresponding to different structural change models and compared them to the experimental DADS. To quantitatively evaluate these comparisons, we employed the reduced chi-square (χ_v^2) criterion. In **Extended Data Figs. 2–4**, it can be found that the best-fit models (highlighted with yellow shades) exhibit the smallest value of χ_v^2 , whereas the models whose simulated curves deviate from the experimental ones show relatively large χ_v^2 . This intuitively validates the usage of reduced chi-square. The application of the reduced chi-square criterion for comparing theoretical and experimental results is well-established and widely utilized. We cited relevant references supporting this approach.

P.10 line 279 states that 'typical' pump-probe experiments only determine dynamics of photoexcited molecules. While the observation of non-photoreacted molecules is indeed a strength of this article, a reference to other 'untypical' pump-probe experiments observing solvation cage dynamics or ground state dynamics could better contextualize the results of this study.

→ We appreciate the reviewer for understanding the key points we aimed to highlight and for offering valuable suggestions for this study. In response, we searched for several previous pump-probe studies that could be regarded as 'untypical', such as the observation of solvent reorientation in optical Kerr effect [*J. Am. Chem. Soc.* **143**, 14261–14273 (2021)] as well as the rotational dephasing of depleted ground state molecules [*Nat. Commun.* **13**, 522 (2022)]. However, we note that the 'untypical' aspects of these studies differ from those observed in our work. Specifically, our study reveals the molecular structural changes in the solute while the aforementioned studies primarily focus on solvents or bulk orientational changes without molecular structural transformations. Given these distinctions, we found it challenging to find relevant references directly comparable to our study. Hence, we kindly ask for the reviewer's understanding in not providing additional citations to the sentence, as doing so might incur unintended confusion to the readers.

In Figure 4a, the rotation of C₂F₄I from anti to gauche is represented as occurring through the T1 state with a height of 3.4 kcal/mol. How can this process happen in 1.2 ps, while the transition through the 4.0 kcal/mol T1 state of the gauche C₂F₄I₂ molecule towards the anti state takes 26 ps?*

→ According to the Arrhenius equation ($k = Ae^{-E_a/kT}$), the reaction rate (k) is determined by not only the activation energy (E_a) but also the pre-exponential factor (A). Here, A is affected by various factors, such as the rotational, vibrational, and translational states of the molecule. For example, the rotational moments of inertia of each molecule affect the relative rotational partition function between the ground state (either *anti* or *gauche*) and the transition state. The presence of one more heavy atom in C₂F₄I₂ compared to C₂F₄I• restricts rotational freedom at the TS. This leads to a smaller $Z_{\text{rot}}^{\text{TS}}/Z_{\text{rot}}^{\text{GS}}$ in C₂F₄I₂ compared to in C₂F₄I•, reducing the pre-exponential factor and slowing down the reaction. The shape and stiffness of the TS also plays a role since a stiffer TS only allows a restricted number of modes to

trespass the TS, resulting in smaller $Z^{\text{TS}}_{\text{vib}}/Z^{\text{GS}}_{\text{vib}}$, again reducing A and slowing down the reaction.

The nature of rotationally isomerizing molecules is one other reason for such deviation. While the *gauche*-to-*anti* isomerization of $\text{C}_2\text{F}_4\text{I}_2$ occurs between ground state molecules, the *anti*-to-*gauche* isomerization of $\text{C}_2\text{F}_4\text{I}\cdot$ occurs between the transiently formed radical intermediates that partially retain the absorbed photon energy. As the 267 nm photon carries the energy of 448 kJ/mol and the primary I dissociation requires ~ 270 kJ/mol, the remaining several hundreds of kilojoules will remain at $\text{C}_2\text{F}_4\text{I}\cdot$ and $\text{I}\cdot$. These energies are then consumed (1) to fire the dissociating iodine from the parent and (2) to form vibrationally hot states that will eventually dissipate heat to the solvents, but also (3) to overcome the rotational activation barrier easier than those starting from the ground state as well. Therefore, the rotational isomerization in $\text{C}_2\text{F}_4\text{I}\cdot$ could occur faster than what can be calculated based on the energy the ground state *anti* needs to overcome the activation barrier. We note that all the effects mentioned in these descriptions do not affect the $e^{-E_a/kT}$ term and are only present in the pre-exponential factor, A.

We included the following discussion in the “Rotational isomerization along C–C and C–C• bonds” subsection of “Results and discussion” section in the main text.

“In general, the observed rotational isomerization time constants of 1.2 ps and 26 ps are comparable to those reported in several previous studies, which range from a few picoseconds to tens of picoseconds. However, we note that the time constant of 1.2 ps for the C–C• bond is drastically faster than the time constant of 26 ps for the C–C bond. This discrepancy may arise from different distributions of vibrational and rotational states contributing distinctively to the pre-exponential factor, and the energetically hot nature of $\text{C}_2\text{F}_4\text{I}\cdot$ (see section 7 of SI for details).”

We also added the following discussion to section 7-1 of SI.

“7. Discussions on the rotational isomerization dynamics

7-1. Qualitative assessment of the time constants

The time constant of rotational isomerization observed in this study is comparable to those reported in a number of previous studies ranging from several picoseconds to tens of picoseconds. However, while the time constant of 26 ps for the C–C bond is typical, the time constant of 1.2 ps for the C–C• bond is slightly faster than what is generally expected for single bond rotation. This observation can be understood by considering the nature of rotationally isomerizing molecules.

In C₂F₄I₂, the *gauche*-to-*anti* isomerization occurs between molecules in the ground state. Conversely, the *anti*-to-*gauche* isomerization of C₂F₄I• occurs between transiently formed radicals that partially retain the absorbed photon energy. As the 267 nm photon carries the energy of 448 kJ/mol and the primary I dissociation requires ~270 kJ/mol, the remaining energies of several hundreds of kilojoules remains within C₂F₄I• and I•. This residual energy is then used to (1) propel the dissociating iodine from the parent molecule, (2) form vibrationally hot states that eventually dissipate heat to the solvents, and (3) overcome the rotational activation barrier more readily than those starting from the ground state. Here, part (3) could make the rotational isomerization in C₂F₄I• to occur faster than what can be predicted solely on the activation energy required for the ground state *anti*-conformers.

Meanwhile, the reaction rate (*k*) is determined not only by the activation energy (*E_a*) but also by the pre-exponential factor (*A*). The pre-exponential factor depends on various properties, such as the rotational, vibrational, and translational states of a molecule. For example, the rotational moments of inertia of each molecule affect the relative ratio of rotational partition functions between the ground state (either *anti* or *gauche*) and the transition state (TS). The presence of an additional heavy atom in C₂F₄I₂, compared to C₂F₄I•, restricts rotational freedom at the TS. This causes a smaller $Z_{\text{rot}}^{\text{TS}}/Z_{\text{rot}}^{\text{GS}}$ for C₂F₄I₂ compared to C₂F₄I•, thereby reducing the pre-exponential factor and slowing down the reaction. The shape and stiffness of the TS also plays a role since a stiffer TS allows a limited number of modes to trespass the TS, resulting in a smaller $Z_{\text{vib}}^{\text{TS}}/Z_{\text{vib}}^{\text{GS}}$, again reducing *A* and slowing down the reaction.

We emphasize that all the effects described here influence only the pre-exponential factor, *A*, without affecting the $e^{-E_a/kT}$ term. This explains why the difference in activation barriers (4.0 kcal/mol for C₂F₄I₂ and 3.2 kcal/mol for C₂F₄I•) is less pronounced compared to the difference in time constants (1.2 ps and 26 ps), while the RRKM theory yields the values far closer to the experiment (1.15 ps and 20.2 ps).”

P.24 in the main text and p.18 in the SI seem to contradict each other. Please clarify if the rotational isomerization takes place through the T1 or the T2 transition state. Are the experimentally extracted values shown in Table 1 or Table 2?

→ We thank the reviewer for identifying the typo. We reviewed the usage of TS1 and TS2 so that their definitions (TS1 at 0° and TS2 at ~120°) remain consistent throughout the text.

The experimentally extracted values are listed in Tables 1 and 2 as well as Extended Data Table 1. Specifically, the experimentally extracted fraction parameters are written in the 2nd row (“fs-TRXL”) of Table 1 and compared with those from the previous ps-TRXL study. The experimentally extracted time constants are shown in the 1st row (“Experiment”) of Table 2 and compared with those estimated through theoretical estimations. Finally, the experimentally obtained structural parameters are summarized in the 3rd row (“fs-TRXL”) of Extended Data Table 1 and compared with theoretically computed values (1st row) and those from the previous ps-TRXL study (2nd row). Therefore, all major quantitative conclusions in this study are summarized in these three tables.

Detailed comments:

The last row in Table 1 is split into two lines, separating the error bars into two parts. Please correct the formatting.

→ We thank the reviewer for the suggestion. We corrected the formatting to enhance the readability.

P.6 ll. 137-138, From context, the meaning can probably be constructed, but the wording makes it unclear: “... the time-independent fingerprints of each structural dynamics and their temporal evolutions, respectively”. I.e.: If the fingerprints are time-independent, how do they show a time evolution? Clearer wording might help the reader.

→ In this sentence, “time-independent fingerprints” part is designated to explain the LSVs, while “their temporal evolutions” part is for the RSVs. Nevertheless, since we recognized that the sentence could cause confusion, we modified the sentence structure with a little more details to clarify the meaning as follows.

“LSV(q) contains the representative signals in q -space that encode the structural information of each chemical species, RSV(t) describes their temporal evolution, and S quantifies the relative contribution of each rank to the data.”

*In the graphical abstract figure, the red vertical line represents the photoexcitation. This makes sense in the top part of the figure. However, the current version implies that anti C2F4I**

and gauche C2F4I2 get excited again before starting to rotate (in the center and bottom boxes). Please clarify this representation.

→ We understand and appreciate the point made by the reviewer. We modified the graphical abstract addressing the issue as follows. Specifically, we removed the photoexcitation marks from the second and third rows and added the greenish color at the beginning of the second row to express that the C–C• rotation takes place from the product of the C–I dissociation.

The shading of the 'unknown' region in Figure 1 reduces the clarity and interpretability of the figure.

→ Since the drawings inside the shaded region of **Figure 1** reflect the dynamics revealed in this study, we made them opaque to avoid the impression that this is already known or generally accepted. Still, we agree with the reviewer's comment on interpretability and adjusted the shading as follows.

Additionally, the figure caption describes the gauche conformers as “yellow” in line 303, while the color in a printed as well as the online version of the manuscript is green.

→ We thank the reviewer for identifying the typo, which we have corrected in the revised manuscript.

The data in Figure 2 appears very crowded at early time points, impeding the identification of the time constant at 1.2 ps. Please consider revising the figure to improve clarity.

→ We appreciate the reviewer’s observation regarding the crowded appearance of the data at early time points in Figure 2. To address this concern, we have adjusted the linear domain of the t -axis from 5 ps to 3.5 ps in the figure to improve clarity. To keep the consistency of t -axis among the SVD-related figures, we also revised Supplementary Figure 4 as well. The revised figures are as follows:

P.8, the referenced Figure S15 does not exist.

→ Figure S15 is a typo of Extended Data Fig. 8, which includes the PEC along the dihedral angle. We appreciate the reviewer for identifying the typo, which we have corrected in the revised manuscript.

P.4, the numbers 34 and 35 in the citation are not separated by comma.

→ We thank the reviewer for identifying the typo, which we have corrected in the revised manuscript.

Responses to the comments from Reviewer #2

→ The authors appreciate the reviewer for considering our manuscript.

Responses to the comments from Reviewer #3

The internal bond rotation in molecules plays a crucial role in determining reactivity, stereochemistry, and the outcomes of reactions. As a result, the associated dynamics have been extensively studied both theoretically and experimentally. However, an atomic-level understanding of the structural dynamics of ultrafast single-bond rotation has remained elusive, primarily due to the challenge of achieving both an ultrafast timescale and the molecular structure resolution required for such studies. This manuscript addresses this challenge by reporting recent advances in directly probing the photo-induced ultrafast structural dynamics of carbon–carbon single-bond rotation in transient species of photoexcited 1,2-tetrafluorodiiodoethane ($C_2F_4I_2$) using femtosecond time-resolved X-ray liquidography (TRXL).

In the experiment, a highly non-equilibrium state was induced by an fs UV pump pulse, resulting in a distribution of anti- and gauche-conformers of the tetrafluoroiodoethyl radical ($C_2F_4I^\bullet$) and 1,2-tetrafluorodiiodoethane. The population evolution of these transient species, as they relax back to thermal equilibrium, was monitored in real time using TRXL, which allowed the conformer ratios and structures of each reacting species to be captured. Combined with extensive theoretical simulations, the study reveals that the rotational

isomerization of $C_2F_4I^\bullet$ and $C_2F_4I_2$ follows *anti-to-gauche* and *gauche-to-anti* pathways with time constants of 1.2 ps and 26 ps, respectively. For the first time, this work provides atomic-level insight into the kinetics and structural dynamics of single-bond rotation.

Overall, this is a cutting-edge, solid, and thorough study, suitable for publication in a high-profile journal.

→ The authors sincerely appreciate the positive feedback on the significance of our findings.

However, I have a few points and questions that should be addressed before final acceptance:

(1) In lines 231 to 232 on page 9, should “the depletions of interatomic distances in *anti-C₂F₄I₂* and the formations in *gauche-C₂F₄I₂*” be revised to “the depletion of *gauche* and the formation of *anti-C₂F₄I₂*” instead?

→ We thank the reviewer for identifying the typo, which we have corrected in the revised manuscript.

(2) The current experiments were conducted under a 267-nm pump light. How would the branching ratios, rotation time constants, and structures of the transient species change with varying pump photon energy?

→ One previous study on $C_2F_4I_2$ [*Phys. Chem. Chem. Phys.* **20**, 12650 (2018)] suggests that the *anti-to-gauche* ratios of initially photoexcited and depleted (i.e., those converted to radicals) molecules depend on the excitation wavelength. According to this study, *anti-C₂F₄I₂* preferentially absorbs longer wavelengths compared to *gauche-C₂F₄I₂*. Thus, using longer wavelengths than 267 nm would deplete more *anti-C₂F₄I₂* and generate more *anti-C₂F₄I[•]* within their respective *anti/gauche* mixture. This would increase the perturbation from dynamic equilibrium for both $C_2F_4I_2$ and $C_2F_4I^\bullet$ by pulling the branching ratio in favor of *anti*, intensifying the rotational isomerization. Conversely, using shorter wavelengths would have the opposite effect, pulling the branching ratio in favor of *gauche*.

About the time constants, we expect that the rotational time constants will not depend much on photon energy in general since the potential energy curve around the ICCI (or ICC[•]) dihedral angle depends solely on the properties of a molecule. Especially, the *gauche-to-anti* rotation of $C_2F_4I_2$ will not be affected since the process happens only through the

concentration gradient where the molecules stay non-photoexcited. However, the *anti-to-gauche* rotational time constant of $C_2F_4I\bullet$ may slightly vary depending on which rotational or vibrational eigenstates the photon activates and how much energy is remaining at $C_2F_4I\bullet$ after the primary dissociation. Finally, the reported *anti* and *gauche* conformer structures of $C_2F_4I_2$ and $C_2F_4I\bullet$ will not change, as they are only determined by the potential energy landscape of the molecule.

Additionally, how might the rotation dynamics differ in gas and liquid phases? In my opinion, addressing these questions would provide a broader perspective and further strengthen the paper.

→ The key distinction between solution (liquid) and gas phases comes from the influences of the solvent shell and its interactions with the solute during the dynamics. First of all, the solvent molecules around the solute can stabilize the solute through the process of solvation. The degree of stabilization varies for reactants, intermediates, and products, depending on their specific interactions with the solvent. In the liquid phase, solvation can preferentially stabilize certain states over others—such as polar intermediates or products—resulting in changes to their relative stabilities. This differential stabilization affects the equilibrium between reactants, intermediates, and products, thereby influencing the excitation and rotational isomerization ratios. However, considering the non-polar and aprotic nature of cyclohexane solvent, the explicit interaction between $C_2F_4I_2$ and the cyclohexane solvent is expected to be minimal. Therefore, it is anticipated that the potential energy surface of $C_2F_4I_2$ in solution will not differ significantly from that of $C_2F_4I_2$ in the gas phase in the ground state.

Still, we cannot exclude the possibility of interactions between the iodine radical and solvent molecules. These interactions could alter the ultrafast radical rotation, which occurs within the timescale of a few hundred femtoseconds to a few picoseconds. The presence of the I atom near the radical may also exert a lasting effect on isomerization. However, we kindly ask for the reviewer's understanding in not including these discussions in detail, as we are preparing a separate paper that extensively focuses on them.

In addition, the presence of the solvent shell in solutions also affects the experimental sensitivity on the structural dynamics of rotational isomerization. In particular, the *anti* and *gauche* conformers of $C_2F_4I\bullet$ only differ by the relative position of fluorine atoms, making them

hard to differentiate only by the Debye equation (the scattering signals only from the solute). However, the solvent shell around the two conformers have different structures due to their distinct partial charges and conformations, yielding differentiable cage scattering signals (the scattering signals from the solute-solvent structures) between *anti* and *gauche* conformers.

As for the comparison, we are aware of a number of ultrafast gas-phase experiments, such as *Phys. Rev. A* **100**, 023402 (2019). However, based on our understanding, these studies have primarily focused on the coherent C–I bond dissociation and not discussed isomerization or concentration changes between conformers. It is likely due to the aforementioned differences in experimental environments, as our choice of solution environment has advantages to capture such subtle structural (in case of $C_2F_4I^\bullet$, with the aid of cage scattering signals) and concentration (in case of $C_2F_4I_2$, due to the high S/N ratio provided by fs-TRXL at XFELs) changes.

We added the following discussion regarding this point to section 7-2 of SI.

“7. Discussions on the rotational isomerization dynamics

7-2. Comparative discussions on the rotational isomerization in the gas phase

We note that a number of previous time-resolved studies investigating the dynamics of $C_2F_4I_2$ and other similar haloalkanes have not captured rotational isomerization, despite utilizing scattering or diffraction techniques that are directly sensitive to structures. We propose that the key distinction lies in the environment, considering that the previous femtosecond time-resolved studies measured $C_2F_4I_2$ in the gas phase while our study probed $C_2F_4I_2$ dissolved in a solvent. Unlike in the gas phase, where the solute can move freely via Brownian motions, the solution environment confines the solute within the solvent shell. This solvent shell and its interactions with the solute exert a critical influence on not only the dynamics itself but also the sensitivity of each species on the measured signals.

The impact of the solvation shell on the dynamics primarily arises from its stabilizing effect. Here, the degree of stabilization varies for reactants, intermediates, and products, depending on their specific interactions with the solvent. In other words, solvation can preferentially stabilize certain states over the others, modifying the free energy differences and activation barriers, thereby shifting the kinetics and equilibrium ratios between reactants, intermediates, and products. Since cyclohexane is nonpolar and aprotic, it can stabilize (1) the nonpolar solutes (*anti*- $C_2F_4I_2$) only via dispersion forces and (2) the polar solutes additionally via dipole-induced dipole interactions. The stabilization effect is slightly stronger for the species at the TS, as they are more polar than the *anti* and *gauche* conformers.

This implies that the activation energy is slightly reduced (and the kinetic rate is slightly increased) in cyclohexane compared to the gas phase. However, the overall stabilization effect would be minimal as dispersion and dipole-induced dipole interactions available in cyclohexane are weak. In contrast, we expect the stabilization effect to be significantly more pronounced in polar solvents, leading to more drastic changes in reaction kinetics compared to the gas phase.

Furthermore, the presence of a solvent shell enhances the sensitivity of scattering data to the structural dynamics of rotational isomerization. For instance, the *anti* and *gauche* isomers of C₂F₄I• only differ by the relative positions of the fluorine atoms, making them less differentiable based solely on the scattering signals from the solute, as in the case in gas-phase studies. In contrast, in solution, the solvent shell surrounding the two conformers have different orientations due to variations in partial charge distribution and molecular conformation. The difference in solvation environments renders the scattering signals of the *anti* and *gauche* conformers differentiable, enabling reliable retrieval of the *anti-to-gauche* isomerization dynamics for C₂F₄I•.”

Responses to the comments from Reviewer #4

This is a very interesting manuscript on the ultrafast photoinduced reactions of 1,2-tetrafluorodiodoethane (C₂F₄I₂) in solution as visualized with femtosecond solution X-ray scattering. The studied molecule features a C-C single bond that allows “the two ends” to rotate around the connecting C-C bond and the question posed here was whether and in which way photoexcitation of the molecule would make this C-C bond rotate, either in the ensemble of intact (parent) molecules or in the radical photoproduct C₂F₄I after I dissociation. Seeing such rotational isomerization or bond rotation is not easy because they are often masked by other, more drastic changes of structures (such as dissociation) and it is not clear a priori how one can be sensitive to such bond rotations anyway. It is evident, though, that it is essential to probe or visualize and understand bond rotations such as those of C-C bonds as they evidently influence the course of important chemical reactions and biological processes. Any step that brings us closer to visualizing C-C bond rotations in real time to then understand the underlying rotational isomerization reactions is highly impactful in a broad range of areas, I think. The authors found a great trick, I find, to do such a step and to visualize in real time photoinduced C-C bond rotations of a molecule in solution by placing heavy F and I atoms in their molecule and seeing how these move with respect to each other with time-

resolved X-ray scattering. This enabled them to infer the underlying C-C bond rotation dynamics.

The presented investigation builds on an earlier study by part of the same authors with the same method, this time with measurement conducted at a femtosecond X-ray free-electron laser source where the former study was performed at a synchrotron radiation source. Clearly, the improvement in temporal resolution from (order of magnitude) 100 picoseconds (ps) to 100 femtoseconds (fs) enabled the authors of the present investigation to reveal new and very interesting information. This study is also a great illustration of the capabilities of X-ray lasers and the new opportunities for studies of condensed-phase chemical reactions arising there.

The manuscript is very well written with very clear figures and a clear and systematic presentation of the results. The main text and supporting information complement each other very well and allow the reader to easily switch from broad to detailed views. The science is sound, and the data and analysis appear to be technically valid. Overall, the manuscript describes a nice and consistent story. The manuscript, I think, is very accessible, interesting to read and should appeal to a broad readership of scientists interested in seeing, understanding, and imagining chemical reactions or reaction steps in real time.

→ We sincerely thank the reviewer for their favorable assessment of our work and its implications.

Of the two main results and claims of the study, the rotational isomerization between anti and gauche of C2F4I and the rotational isomerization between anti and gauche of C2F4I2, I find those of C2F4I (1.2 ps time constant in the kinetics of the reaction) very well motivated and substantiated. The ones of C2F4I2 (26 ps time constant), I think, are less clear and less validated. I understand that this latter aspect is an exciting given that these are rotational dynamics of the parent (intact non-dissociated) molecule that indirectly relate to the initial photoexcitation. I just find the evidence not very convincing, and I don't think that the authors "need" this aspect or detail of the study to have a great study or story. I thus suggest focusing the manuscript more on the C-C bond rotation of C2F4I and, in turn, weaken the claims on

the C-C bond rotation in C₂F₄I₂. What we are left with is still a wonderful achievement in my view that is worth reporting.

→ We once again thank the reviewer for high evaluation of our findings on C₂F₄I•. That said, we also believe that the evidence is sufficient to support our findings about C₂F₄I₂. In TRXL, the transient scattering intensity depends on two factors: the amplitude of difference scattering curves from a single molecule and the number of molecules (=concentration) undergoing those dynamics. We recognize that the rotationally isomerizing fraction of C₂F₄I₂ (78:22 to 79:21) is far smaller than that of C₂F₄I• (91:9 to 71:29), which might have given an impression that the former is less convincing than the latter. However, the actual number of molecules undergoing rotational isomerization are not much different to each other since the isomerizing concentration of C₂F₄I₂ is 0.3 mM and that of C₂F₄I• is 0.5 mM (**Fig. 4b**). Also, if we compare the amplitude of difference scattering curves from single C₂F₄I₂ and C₂F₄I•, the presence of two I atoms in C₂F₄I₂ makes the rotational isomerization far more pronounced in the scattering signal. This is proven as the exclusion of the time constant of 26 ps attributed to the rotation of C₂F₄I₂ significantly worsens the chi-square criteria in **Extended Data Fig. 5**. The quality-of-fit of the correct model in **Extended Data Fig. 4** (about the time constant of 26 ps assigned to *gauche*-to-*anti* rotational isomerization of C₂F₄I₂) is also far better than those of the other competing models, just like in **Extended Data Fig. 3** (about the time constant of 1.2 ps assigned to *anti*-to-*gauche* rotational isomerization of C₂F₄I•).

I have additional comments that the authors may consider in a potential revision of their manuscript:

I find that the term “radical” in the title fits to the main findings around C-C bond rotations in C₂F₄I and I think that that should be the focus of this manuscript. The abstract though (and the main text and conclusion sections in fact as well) does not express this focus but rather treats C₂F₄I and C₂F₄I₂ seemingly equally.

→ We understand the reviewer’s comment that the dynamics of radical rotation should be highlighted more if it is emphasized in the title. We ask for the reviewer’s understanding, however, that the other reviewer in fact recommended the exact opposite. According to that comment, this study should highlight more on the rotational isomerization of C₂F₄I₂ and the

importance of capturing the dynamics occurring between non-photoreacted molecules. Thus, we decided to introduce additional discussions on the rotational isomerization of $C_2F_4I\bullet$ while retaining the current discussions on the rotational isomerization of $C_2F_4I_2$. Considering these feedbacks, we added a discussion assessing the factors that could influence the kinetics for the C–C• bond rotation in “Rotational isomerization along C–C and C–C• bonds” subsection of “Results and discussion” section in the main text and section 7 “Discussions on the rotational isomerization dynamics” of SI.

In the abstract “However, structural dynamics of ultrafast single-bond rotation ... have remained elusive due to the limited sensitivity of spectroscopic tools in resolving molecular structures.” is a bit misleading or not specific enough in my view. Obviously, there are numerous spectroscopic tools that are very sensitive to (changes of) molecular structures. The bond rotations addressed here, though, may be hard to detect with spectroscopic tools. I suggest reworking this sentence to better express why bond rotations are so hard to probe spectroscopically.

→ As the reviewer correctly pointed out, our original intention in the referred sentence was to highlight the challenges of structurally resolving rotational isomerization when relying solely on spectroscopy. We certainly did not mean to imply that spectroscopic tools are insensitive to molecular structures in general. Accordingly, to avoid such confusion, we have revised the abstract as follows. We highlighted the key differences using bold font. (Please note that other minor changes were made in response to comments from another reviewer and are also reflected in the abstract below.)

“Bond rotation is an important phenomenon governing the fate of reactions. In particular, heterogeneously substituted ethane derivatives provide distinct structural conformations around the bond, empowering them as ideal systems for studying the rotation along carbon-containing single bonds. However, structural dynamics of ultrafast single-bond rotation, especially along C–C• bonds, have remained elusive **as tracking the detailed changes in structural parameters during the rotational isomerization is challenging with conventional spectroscopic tools**. Here, we employed femtosecond time-resolved X-ray liquidography (TRXL) to visualize the rotational isomerization between *anti* and *gauche* conformers of tetrafluoroiodoethyl radical ($C_2F_4I\bullet$) and 1,2-tetrafluorodiiodoethane ($C_2F_4I_2$), simultaneously. The TRXL data captured perturbations in conformer

ratios and structures of each reacting species, revealing that the rotational isomerization of C₂F₄I• and C₂F₄I₂ follows *anti-to-gauche* and *gauche-to-anti* paths with time constants of 1.2 ps and 26 ps, respectively. These findings also align with the computational predictions. This work offers an atomic-level insight into the kinetics and structural dynamics of single-bond rotation.”

In the introduction in “... occurs with time constants ranging from 10 ps to 100 ps at room temperature...” it is not quite clear to me what makes this range so large (depending on what do theoretical studies predict different times?).

→ The discrepancy stems from (1) the computational methods and (2) the chemical environment under which the rotational isomerization of *n*-butane was modeled in these studies. For example, in ref 23 [*Chem. Phys. Lett.* **75**, 162–168 (1980)], the authors comparatively reported the rotational time constants for *n*-butane in rigid matrix (5.8 ps) and in CCl₄ solution (7.1 ps) using viscous continuum model (the model that accounts for the 3D shape of molecules). Meanwhile, in ref 22 [*J. Chem. Phys.* **115**, 7285–7292 (2001)], the rate constant is estimated in pure *n*-butane by computing a microscopic average of statistically generated transition paths. This study reported the value of 60.6 ps for the rotational isomerization and also cited similar values from other references (56.8 ps from the study [*J. Chem. Phys.* **92**, 3062 (1990)] and 52.6 ps from the study [*J. Chem. Phys.* **87**, 5700 (1987)]).

In fact, we found many early computational studies (mostly between 7–80s to 2000s) focusing on the rotational isomerization dynamics of ethane and simple ethane derivatives like *n*-butane. While we had to choose among them due to the citation limit, we found that the time constants reported in these (even uncited) studies mostly fall within the boundary between 10 and 100 ps despite the diversities of computational methods and chemical environments. It is why we wrote the sentence as it currently is. At the same time, we believe that a more detailed review of how these rotational time constants are computed under different conditions is unnecessary for the current study, as we only cited them as historical backgrounds, and our focus is not on these ‘simple derivatives’ but rather on molecules with heavy scatterers. Instead, we concisely added why such discrepancies might occur, and also changed the lower bound from 10 ps to 1 ps considering that one of the two cited references reported the value of 7.1 ps. Consequently, we changed the sentence as follows.

“Early theoretical studies have shown that the rotational isomerization of *n*-butane, one of the simplest ethane derivatives, occurs with time constants ranging from 1 ps to 100 ps at room temperature under varying chemical environments.”

Responses to the comments from Reviewer #1

We appreciate the authors careful work in addressing the points raised in our original review report. The resubmitted work is a remarkable achievement of clear scientific interest and should be published.

→ We thank the reviewer for acknowledging the improvements made during the revision. We once again express our regards for the fruitful discussions given by the reviewer.

Responses to the comments from Reviewer #2

→ The authors appreciate the reviewer for considering our manuscript.

Responses to the comments from Reviewer #3

In this revision, the authors have made adequate modifications to address the concerns raised by the reviewers in the previous round. Overall, the revised manuscript is clearer and more cohesive. It is now ready for publication.

→ We thank the reviewer for evaluating the improvements made during the revision. We once again express our regards for the fruitful discussions given by the reviewer.

Responses to the comments from Reviewer #4

The authors have adequately addressed and resolved my concerns. This relates to the changes made to the manuscript in the revised version, the authors' responses to the other reviewers and the response to my own questions and comments. I find that this is a very interesting and complete study, with a mature technique on an interesting problem, the real-time observation of C-C bond rotation as triggered by a short light pulse, in a great form of a well-written manuscript.

→ We appreciate the reviewer's understanding regarding our responses to the revision step. We once again express our regards for the fruitful discussions given by the reviewer.